# Regional Explanations: Bridging Local and Global Variable Importance

**Salim I. Amoukou**[*]
J.P. Morgan AI Research

**Nicolas J-B. Brunel**
LaMME, ENSIIE, University Paris Saclay & Capgemini Invent

## Abstract

We analyze two widely used local attribution methods, Local Shapley Values and LIME, which aim to quantify the contribution of a feature value $x_i$ to a specific prediction $f(x_1, \ldots, x_p)$. Despite their widespread use, we identify fundamental limitations in their ability to reliably detect locally important features, even under ideal conditions with exact computations and independent features. We argue that a sound local attribution method should not assign importance to features that neither influence the model output (e.g., features with zero coefficients in a linear model) nor exhibit statistical dependence with functionality-relevant features. We demonstrate that both Local SV and LIME violate this fundamental principle. To address this, we propose R-LOCO (Regional Leave Out COvariates), which bridges the gap between local and global explanations and provides more accurate attributions. R-LOCO segments the input space into regions with similar feature importance characteristics. It then applies global attribution methods within these regions, deriving an instance's feature contributions from its regional membership. This approach delivers more faithful local attributions while avoiding local explanation instability and preserving instance-specific detail often lost in global methods.

## 1 Introduction

Machine learning models are valued for their predictive capabilities, but often lack transparency. This opacity poses issues in high-stakes domains such as healthcare, finance, and criminal justice, where understanding the "why" behind a decision is as critical as the decision itself. In response, eXplainable AI (XAI) has emerged, developing methods and tools to explain model predictions.

These tools can be categorized into local and global methods. Local explanations aim to provide insights into individual predictions, while global explanations focus on understanding the overall behavior of a model across the entire input space. Popular local methods, such as *Local* Shapley Values **L-SV** [Lundberg and Lee, 2017b] and **LIME** [Ribeiro et al., 2016b], seek to create a local linear approximation of the model within the vicinity of a given instance. In contrast, global methods primarily consist of "leave-one-out" approaches [Lei et al., 2016, Williamson and Feng, 2020, Covert et al., 2020b, Gan et al., 2022], which assess performance changes when a variable is excluded.

In this paper, we focus on local attribution methods, specifically **L-SV** and **LIME**, which are widely used but have limited theoretical understanding. While the intuition behind these methods are appealing, the exact quantities they estimate remain unclear. Apart from the case of linear or additive models [Bordt and von Luxburg, 2023, Garreau and Luxburg, 2020], there is little work explaining the specific quantities these methods compute. A theoretical study on **LIME**, by Garreau and Luxburg [2020], shows that in the case of a linear model, the **LIME** coefficients are proportional to the partial derivatives. However, it also reveals that the coefficient of important variables can vanish by simply changing a parameter of the method.

---

[*]Correspondence to: Salim I. Amoukou <salim.ibrahimamoukou@jpmorgan.com>

39th Conference on Neural Information Processing Systems (NeurIPS 2025).

Beyond these established concerns, we highlight a fundamental limitation of **L-SV** and **LIME** that is orthogonal to the well-known issue of being "true to the data vs true to the model" [Chen et al., 2020]. Building on our conclusion in Amoukou et al. [2022], we posit that a reliable local attribution should not assign importance to features that neither influence the model output functionally nor exhibit statistical dependence on features that do. Even under ideal conditions—with independent features, exact computations, and no estimation error—we show that both **L-SV** and **LIME** violate this principle by attributing importance to irrelevant features. This highlights an often-overlooked vulnerability in these popular methods, distinct from issues of approximation error or feature correlation.

In contrast, most global methods are supported by the existing literature on feature importance [Breiman, 2001, Lei et al., 2016, Williamson and Feng, 2020] and global sensitivity analysis [Iooss and Lemaitre, 2015], and are backed by strong consistency and inference results. Our goal is to leverage global methods to define more accurate local attributions for each individual while benefiting from the advantages of global methods. We aim to find a partition of the input space where observations in each cell of the partition exhibit the same behavior concerning the underlying model. Then, we associate each observation with the global importance conditional on the cell it belongs to, hence deriving an attribution that is more locally faithful and possesses sound statistical properties.

**Notations.** Consider a dataset represented as $\mathcal{D}_n = \{(\boldsymbol{X}_i, Y_i)\}_{i=1}^n$, where $\boldsymbol{X}_i = (X_{i1}, \ldots, X_{ip}) \in \mathcal{X} \subseteq \mathbb{R}^p$ denotes the input variables and $Y_i \in \mathcal{Y} \subseteq \mathbb{R}$ represents the output, and $(\boldsymbol{X}_i, Y_i)$ are i.i.d. observations of $(\boldsymbol{X}, Y) \sim P = P_{\boldsymbol{X}} P_{Y|\boldsymbol{X}}$. We use $\boldsymbol{X}_S = (X_i)_{i \in S}$ to denote the subset of features, and $[p] = \{1, \ldots, p\}$, and $\mathcal{P}(D)$ represents the power set of a set $D$.

## 2 Limitations of *Local* Shapley Values

A cooperative game is a pair $(D, v)$, where $D = \{X_1, \ldots, X_p\}$ represents a set of $p$ players, and $v : \mathcal{P}(\{1, \ldots, p\}) \to \mathbb{R}$ denotes a value function that assigns a value to every possible coalition of players, reflecting the worth of each group. The value function $v$ is generally assumed to be positive and increasing monotonically [Dubey and Weber, 1977], which means that if $A \subseteq B$, then $v(A) \leq v(B)$. Here, $v(A)$ represents the value of $\boldsymbol{X}_A$. A key concept in the definition of SV is the marginal contribution, denoted as $\Delta_i(S) = v(S \cup i) - v(S)$. The marginal contribution is the improvement of the value of a coalition $S$ when a given player $i$ is added to the coalition. The SV of $X_i$ is the weighted average of the marginal contributions of $X_i$ across all subsets, expressed as:

$$\phi_{X_i} = \sum_{S \subseteq D \setminus \{i\}} w(S) \Delta_i(S) = \frac{1}{p} \sum_{S \subseteq D \setminus \{i\}} \binom{|D| - 1}{|S|}^{-1} [v(S \cup i) - v(S)].$$

We can establish feature importance, by defining the value function. In the global sensitivity literature, a frequently used value function is $v(S) = \mathbb{V}(\mathbb{E}[Y|\boldsymbol{X}_S])/\mathbb{V}(Y)$, which represents the explained variance or the variance of the best approximation of $Y$ given $\boldsymbol{X}_S$. This value function is nonnegative and monotonically increasing, resulting in a positive global importance measure. When features are independent, this SV is closely related to the functional ANOVA decomposition [Efron and Stein, 1981, Hoeffding, 1948] and Sobol indices [Sobol', 1990, Chastaing et al., 2012, Hooker, 2007]. The resulting SV are commonly referred to as Shapley Effects [Owen, 2014, Owen and Prieur, 2017].

In contrast to the global Shapley Values (SV) approach or Shapley Effects, Lundberg and Lee [2017a], Lundberg et al. [2020] adopt the game theory paradigm to explain a specific prediction $f(x_1, \ldots, x_p)$ with players $D = \{X_1 = x_1, \ldots, X_p = x_p\}$ using the value function $v(S) = \mathbb{E}[f(\boldsymbol{x})|\boldsymbol{X}_S = \boldsymbol{x}_S]$ or $v(S) = \mathbb{E}[f(\boldsymbol{x}_S, \boldsymbol{X}_{\bar{S}})]$. Although debates continue over the choice between these two value functions [Heskes et al., 2020, Janzing et al., 2020, Chen et al., 2020], we assume in this work that the variables are independent, making these value functions equivalent. We refer to the resulting Shapley Values in this context as Local Shapley Values (**L-SV**).

A key distinction between the global SV approach (Shapley Effects) and the local approach **L-SV** hinges on the definitions of their respective value functions. While the global value function $v(S) = \mathbb{V}(\mathbb{E}[Y|\boldsymbol{X}_S])/\mathbb{V}(Y)$ serves as an effective measure of the predictive power of variables $\boldsymbol{X}_S$ for the overall model, it is unclear whether the local value function $v(S) = \mathbb{E}[f(\boldsymbol{x})|\boldsymbol{X}_S = \boldsymbol{x}_S]$ genuinely represents the predictive power of $\boldsymbol{X}_S = \boldsymbol{x}_S$ for the specific prediction $f(\boldsymbol{x})$. In the global case, a high value of $v(S) = \mathbb{V}(\mathbb{E}[Y|\boldsymbol{X}_S])/\mathbb{V}(Y)$ indicates a strong predictive power of $\boldsymbol{X}_S$, while the values taken by $v(S) = \mathbb{E}[f(\boldsymbol{x})|\boldsymbol{X}_S = \boldsymbol{x}_S]$ in the local case do not have any intuitive order, i.e. a high or low value of $v(S)$ does not necessarily reflect the importance of $\boldsymbol{X}_S = \boldsymbol{x}_S$ for the prediction $f(\boldsymbol{x})$ in regression problems, for example.

Moreover, the value function of **L-SV**, $v(S) = \mathbb{E}\big[f(\boldsymbol{X})|\boldsymbol{X}_S = \boldsymbol{x}_S\big]$, can be negative and does not satisfy the monotonic property, which may result in negative **L-SV**. Therefore, a cancelling effect can occur, where a variable that influences the decision ends up with a zero or low Shapley Value because the $\Delta_i(S)$ values across all subsets cancel each other out. It is important to note that Shapley Effects do not encounter the issues mentioned above, as they satisfy the non-negativity criterion suggested for feature importance [Johnson and LeBreton, 2004, Grömping, 2007, Feldman, 2005]. In fact, Feldman [2005] emphasized that an importance measure should be positive, as it evaluates the relative information a variable contributes to the model, and information is inherently non-negative.

While the *global* SV can be interpreted as percentages of the output's variance [Idrissi, 2024], the quantities estimated by the *local* SV are less clear. For example, what is the **L-SV** for a particular $X_1 = x_1$ really telling us about its contribution to the **specific** prediction $f(x_1, \ldots, x_p)$?

Lastly, there is no strong justification for calculating contributions across all possible subsets, as some of these subsets might be poor predictors, thus introducing noise into the feature importance. The average performance of a feature across all submodels may not be indicative of the particular performance of that feature in the set of optimal submodels. Additionally, averaging over all subsets tends to reduce the local aspect of the contribution. To demonstrate this, let's assume we have independent variables $\boldsymbol{X} \in \mathbb{R}^p$, and a piece-wise linear predictor $f$ defined as:

$$f(X) = (a_1 X_1 + a_2 X_2)\mathbb{1}_{X_6 \leq 0} + (a_3 X_3 + a_4 X_4)\mathbb{1}_{X_6 > 0}. \tag{1}$$

Even if we choose an observation $\boldsymbol{x} = (x_1, \ldots, x_p)$ such that $x_6 \leq 0$ and the predictor only uses $x_1, x_2$, the **L-SV** of $\phi_{x_3}, \phi_{x_4}$ is not necessarily zero. Using straightforward calculations, we can show that for all $i \in \{3, 4\}$, $\phi_{x_i} = K\Big(a_i(\boldsymbol{x}_i - \mathbb{E}[\boldsymbol{X}_i])\Big)$ where $K \propto \mathbb{P}(X_6 > 0)$ is a constant.

This highlights that the **L-SV** are not purely local measures but also exhibit global influences. This occurs because when calculating the **L-SV** of $X_3 = x_3$ or $X_4 = x_4$, we also consider subsets $S$ that do not contain $X_6$. By marginalizing and changing the sign of $X_6$, we use the other linear model not used for this observation. We can extend the result above to show a similar issue with continuous piece-wise linear function. This class of functions is quite versatile, as it encompasses neural networks with piece-wise linear activations such as ReLU or hard tanh which correspond to $\max(0, x)$ and $\max(-1, \min(1, x))$ respectively. Indeed, we can view feedforward neural networks as piece-wise linear functions that divide the input space into multiple linear regions, where the network itself behaves as an affine function within each region [Pascanu et al., 2013, Hanin and Rolnick, 2019a,b, Chen et al., 2022].

The important local variables of this model correspond to the coefficients of the linear model associated with the region $A_k$ to which the observation belongs. However, in the following, we provide a result demonstrating that these explanations cannot be retrieved using *Local* Shapley Values.

**Theorem 2.1** (Local Shapley Values for Piecewise Linear Models). *Let $f : \mathcal{X} \to \mathbb{R}$ be a piece-wise linear function defined on a feature space $\mathcal{X} \subseteq \mathbb{R}^p$, which is partitioned into $m$ disjoint hyperrectangles, $\{A_k\}_{k=1}^m$, where each $A_k = \bigotimes_{i=1}^p [l_{i,k}, r_{i,k}]$ with $l_{i,k}, r_{i,k} \in \mathbb{R} \cup \{-\infty, \infty\}$. On each region $A_k$, the function is defined by a linear model $f_k(\mathbf{X}) = \sum_{i=1}^p a_{i,k} X_i + b_k$. The complete function is thus $f(\mathbf{X}) = \sum_{k=1}^m f_k(\mathbf{X})\mathbb{1}_{\mathbf{X} \in A_k}$. Consider an observation $\mathbf{x} = (x_1, \ldots, x_p)$ located in a specific region $A_{k^\star}$ for some $k^\star \in \{1, \ldots, m\}$. The model's prediction is therefore determined solely by the local model $f_{k^\star}$, i.e., $f(\mathbf{x}) = f_{k^\star}(\mathbf{x})$, and the set of truly local active features is $J_{k^\star} = \{i \in [p] \mid a_{i,k^\star} \neq 0\}$. We also define the set of features important globally $G = \{i \in [p] \mid \exists k \text{ s.t. } A_{i,k} \neq \mathbb{R}\}$. Let the covariates $X_i$ be mutually independent. The Local Shapley Value (SV) for the feature-value $X_l = x_l$ is given by $\phi_{x_l} = \sum_{k=1}^m \phi_{x_l}^k$, where each component $\phi_{x_l}^k$ is:*

$$\phi_{x_l}^k = \left(\frac{\mathbb{1}_{x_l \in A_{l,k}}}{\mathbb{P}(X_l \in A_{l,k})} - 1\right) \sum_{S \subseteq D\backslash\{l\}} w(S) v_k(S)$$

$$+ a_{l,k}\left(x_l - \frac{\mathbb{E}\big[X_l \mathbb{1}_{X_l \in A_{l,k}}\big]}{\mathbb{P}(X_l \in A_{l,k})}\right) \sum_{S \subseteq D\backslash\{l\}} w(S) \prod_{i \in S\cup\{l\}} \mathbb{1}_{x_i \in A_{i,k}} \prod_{j \in D\backslash(S\cup\{l\})} \mathbb{P}(X_j \in A_{j,k})$$

$$\tag{2}$$

*with $w(S) = \frac{1}{p}\binom{p-1}{|S|}^{-1}$, and $v_k(S) = \mathbb{E}[f_k(\mathbf{X})\mathbb{1}_{\mathbf{X} \in A_k} \mid \mathbf{X}_S = \mathbf{x}_S]$. The key implication of Equation (2) is that Local SVs can assign importance to a feature $j$ even if that feature is locally irrelevant (i.e., $j \notin J_{k^\star}$) and globally irrelevant (i.e., $j \notin G$).*

See the proof in the Appendix A. Theorem 2.1 demonstrated that SV also present difficulties in expressing local importance measures for neural networks with piece-wise linear activation layers and, more generally, for continuous piece-wise linear functions.

## 3   Limitations of LIME

The core idea of **LIME** [Ribeiro et al., 2016b] is to approximate a complex model with a simpler, interpretable model (e.g., a linear model) in the neighborhood of an instance $x^\star$. However, **LIME** has several limitations due to its dependence on heuristic choices, such as the sampling distribution and the definition of locality. The typical sampling distribution ignores feature dependencies, which can lead to inconsistent explanations. Moreover, defining the neighborhood $\pi_{x^\star}^h$ (e.g., using a Gaussian kernel) and tuning the kernel width $h$ is challenging, particularly in high-dimensional settings, which affects the stability and reliability of the explanations. **LIME** also exhibits instability, with different runs producing varying results due to the randomness in synthetic sample generation. Several studies [Zafar and Khan, 2019, Visani et al., 2020, Zhang et al., 2019, Alvarez-Melis and Jaakkola, 2018, Zhou et al., 2021] highlight these issues. Additionally, we have also demonstrated that **LIME**, like **L-SV**, struggles to detect the locally important variables of the piece-wise model as in (1). For more detailed analysis and further empirical analysis on these limitations, including sensitivity to bandwidth and reproducibility issues, please refer to the Appendix D.

## 4   From Global Explanations to Regional Explanations

In this section, we follow the presentation of Williamson et al. [2021], which introduces a general framework for global variable importance. We consider a comprehensive class $\mathcal{F}$ of functions mapping from $\mathcal{X}$ to $\mathcal{Y}$, and $\mathcal{F}_{-j}$ be the subset of $\mathcal{F}$ containing all functions that disregard the variable $X_j$. The conformity score $V(f(\boldsymbol{X}), Y)$ assesses the predictiveness of a prediction function $f \in \mathcal{F}$ on the observation $(\boldsymbol{X}, Y)$, where a high value implies high predictiveness. We define the oracle predictor with respect to the conformity score $V$ and distribution $P = P_{\boldsymbol{X}} P_{Y|\boldsymbol{X}}$ as follows:

$$f_0 = \arg \max_{f \in \mathcal{F}} \mathbb{E}_P \big[ V(f(\boldsymbol{X}), Y) \big].$$

In a similar manner, we define $f_{-j}$ as the function maximizing $\mathbb{E}_P[V(f(\boldsymbol{X}), Y)]$ over all $f \in \mathcal{F}_{-j}$. Subsequently, we define the *population-level importance* for the variable $X_j$ as the decrease in predictiveness when excluding $X_j$ from $\boldsymbol{X} = (X_1, \ldots, X_p)$. This is commonly referred to as the Leave Out COvariates (**LOCO**) importance in existing literature [Lei et al., 2016, Verdinelli and Wasserman, 2024]. The **LOCO** importance for $X_j$ is defined as:

$$\Psi_j(P) = \mathbb{E}_P \big[ \Delta_j(\boldsymbol{X}, Y) \big] \quad \text{where} \quad \Delta_j(\boldsymbol{X}, Y) = V(f_0(\boldsymbol{X}), Y) - V(f_{-j}(\boldsymbol{X}), Y).$$

We can use any conformity score to measure variable importance, depending on the problem. For regression tasks, we can use $V(f(\boldsymbol{X}), Y) = 1 - [Y - f(\boldsymbol{X})]^2 / \sigma^2$, where $\sigma^2 = \mathbb{E}_P \big[ Y - \mathbb{E}_P[Y] \big]^2$ represents the variance of the target variable $Y$. This conformity score corresponds to the traditional $R^2$ score at the population level, i.e., $R^2 = \mathbb{E}_P[V(f(\boldsymbol{X}), Y)]$. Alternatively, for binary classification problems, we can use $V(f(\boldsymbol{X}), Y) = \mathbb{1}_{Y = f(\boldsymbol{X})}$, which corresponds to the accuracy score at the population level. Regarding the choice of the conformity score, there is no ground truth for variable importance as there are multiple definitions of what makes a variable important [Hooker et al., 2019, Hama et al., 2022, Verdinelli and Wasserman, 2024]. Consequently, the choice of a conformity score should be contingent upon the specific context and goals of the analysis.

Our approach aims to derive local explanations for a specific observation $(\boldsymbol{X}, Y)$ from the population-level importance $\Psi_j(P)$. This involves identifying a partition $\cup_i A_i = \mathcal{X}$ containing observations with similar explanations, which means $\mathbb{V}_P(\Delta_j(\boldsymbol{X}, Y) \mid \boldsymbol{X} \in A_i) \approx 0$ for all $j \in \{1, \ldots, p\}$ simultaneously. In other words, the observations within each partition have low variance in their feature importance. As a result, we can use $\Psi_j(P_{A_i}) = \mathbb{E}_{P_{A_i}} \big[ \Delta_j(\boldsymbol{X}, Y) \big] = \mathbb{E}_P \big[ \Delta_j(\boldsymbol{X}, Y) \mid \boldsymbol{X} \in A_i \big]$ as local explanations for all observations in $A_i$, since the random variable $\Delta_j(\boldsymbol{X}, Y)$ exhibits low variance within $A_i$. Essentially, our approach involves identifying a homogeneous group with respect to importance measure $\Delta_j(\boldsymbol{X}, Y)$ and attributing the global importance of this group as the local explanations of its members. Hence, it permits us to have local explanations while benefiting from all the inference results available for global explanations.

## 4.1 Estimation and inference

To compute our local explanations, we need to compute two quantities: the **LOCO** importance $\Psi_j(P) = \mathbb{E}_P[\Delta_j(\boldsymbol{X}, Y)]$ for any distribution $P$ and the partition $\cup_i A_i = \mathcal{X}$ that group observations by their feature importance similarity. The former has been extensively studied in [Williamson et al., 2021], where the authors proposed a nonparametric efficient estimation procedure using the following plug-in estimator:

$$\widehat{\Psi}_j(\widehat{P}) = \mathbb{E}_{\widehat{P}}[\widehat{\Delta}_j(\boldsymbol{X}, Y)] \quad \text{where} \quad \widehat{\Delta}_j(\boldsymbol{X}, Y) = V\left(\widehat{f}_0(\boldsymbol{X}), Y\right) - V\left(\widehat{f}_{-j}(\boldsymbol{X}), Y\right), \quad (3)$$

$\widehat{P} = 1/n \sum_{i=1}^n \delta_{(\boldsymbol{X}_i, Y_i)}$ is the empirical distribution of $P$ based on $\mathcal{D}_n = \{(\boldsymbol{X}_i, Y_i)\}_{i=1}^n$ and $\widehat{f}_0, \widehat{f}_{-j}$ are estimators of the population minimizers $f_0$ and $f_{-j}$ respectively. $\widehat{f}_0, \widehat{f}_{-j}$ are obtained by building a predictive model for $Y$ using all features in $\boldsymbol{X}$ and after removing the variable $X_j$ respectively. This can be achieved using any machine learning algorithm. Williamson et al. [2021] demonstrated that the estimator defined in Equation (3), is asymptotically efficient and enables valid statistical inference under regularity conditions.

Having obtained a consistent estimator for $\Psi_j(P)$, we now propose a method to derive the partition $\cup_i A_i = \mathcal{X}$. The initial step involves creating a new representation for each observation $\boldsymbol{X}_i = (X_{i1}, \ldots, X_{ip})$ in $\mathcal{D}_n$ using the conformity score $V$. This is expressed as $\tilde{\boldsymbol{X}}_i = \left(\widehat{\Delta}_1(\boldsymbol{X}_i, Y_i), \ldots, \widehat{\Delta}_p(\boldsymbol{X}_i, Y_i)\right)$. Representing the data in the space of feature importance, rather than the original covariate space, allows for the grouping of observations that exhibit consistent behavior with respect to the importance measure $\widehat{\Delta}_j(\boldsymbol{X}, Y) = V(\widehat{f}_0(\boldsymbol{X}), Y) - V(\widehat{f}_{-j}(\boldsymbol{X}), Y)$ across all variables $j \in \{1, \ldots, p\}$ simultaneously.

The subsequent step involves clustering similar observations based on their new representations $\tilde{\boldsymbol{X}}$. This can be achieved using any clustering algorithm, such as K-means [Macqueen, 1967], DBSCAN [Ester et al., 1996], Tree-based clustering approach [Maimon and Rokach, 2014] or Affinity Propagation [Frey and Dueck, 2007], ultimately resulting in a partition of $\mathcal{D}_n$ into $K$ sets $\mathbf{C} = \{C_1, \ldots, C_K\}$. Note that clustering in the feature importance space does not necessarily imply that observations will be closer in the input space. In the experimental section, we demonstrate that clustering in the feature-importance space provides a more meaningful basis for capturing the model's actual behaviour, and we offer a theoretical explanation in Section 5.4.1 within a controlled setting.

The final step entails using the identified clusters $\mathbf{C}$ to define a partition of $\mathcal{X}$. This is accomplished by assigning any new points $\boldsymbol{x} \in \mathcal{X}$ to their nearest partition $C_k$ with respect to a similarity function $d : \mathcal{X} \times \mathcal{X} \to \mathbb{R}^+$. We can use any similarity function. We use Euclidean or Manhattan distance. More formally, we define the corresponding region $\widehat{A}_k$ of cluster $C_k$ as

$$\left\{\boldsymbol{x} \in \mathcal{X} : \sum_{\boldsymbol{X}_i \in C_k} d(x, \boldsymbol{X}_i) < \sum_{\boldsymbol{X}_i \in C_l} d(x, \boldsymbol{X}_i) \text{ for all } k \neq l\right\}.$$

In order to establish a partition of $\mathcal{X}$, we must also account for the set of "undecidable" observations, which are those that simultaneously belong to multiple groups. We define this set $\widehat{A}_{K+1}$ as $\left\{\boldsymbol{x} \in \mathcal{X} : \exists k, l \in [k], \sum_{\boldsymbol{X}_i \in C_k} d(x, \boldsymbol{X}_i) = \sum_{\boldsymbol{X}_i \in C_l} d(x, \boldsymbol{X}_i)\right\}$. Hence, the local explanations of a given observation $\boldsymbol{X}_i$ that belongs to the region $\widehat{A}_k$ can be represented by the vector $\left(\widehat{\Psi}_1(\widehat{P}_{\widehat{A}_k}), \ldots, \widehat{\Psi}_p(\widehat{P}_{\widehat{A}_k})\right)$, where the contribution of the feature $X_j$, $\widehat{\Psi}_j(\widehat{P}_{\widehat{A}_k})$, is defined as

$$\widehat{\Psi}_j(\widehat{P}_{\widehat{A}_k}) = \mathbb{E}_{\widehat{P}_{\widehat{A}_k}}[\widehat{\Delta}_j(\boldsymbol{X}, Y)] = \mathbb{E}_{\widehat{P}}[\widehat{\Delta}_j(\boldsymbol{X}, Y) \mid \boldsymbol{X} \in \widehat{A}_k] = \sum_{(\boldsymbol{X}_i, Y_i) \in C_k} \frac{\widehat{\Delta}_j(\boldsymbol{X}_i, Y_i)}{|C_k|}.$$

We refer to this approach as **R-LOCO**, and details can be found in Figure 6. R-LOCO's construction addresses key limitations of **L-SV**, particularly for models like piecewise linear functions.

**Theorem 4.1.** *Let $f$ be the piecewise linear function from Theorem 2.1. When the **R-LOCO** clustering algorithm identifies cluster $C_{k^\star}$ within regions $A_{k^\star}$, the R-LOCO attributions for $\boldsymbol{x} = (x_1, \ldots, x_p) \in A_{k^\star}$, with $k^\star \in \{1, \ldots, m\}$, are solely influenced by $f_{k^\star}$. They do not depend on coefficients of any $a_{i,k'}$ from other regions $A_{k'}$, where $k' \neq k^\star$.*

Unlike **L-SV**, which can be influenced by inactive components even under ideal conditions (§2), Theorem 4.1 shows that **R-LOCO** assigns importance exclusively to the active region whenever $C_{k^\star} \subseteq A_{k^\star}$. See the proof in §B. In §B.1, we further analyze non-homogeneous (mixed) regions, where the attributions can depend on contamination from other regions; moreover, the induced bias scales proportionally with the contamination level (e.g., with the fraction of the mass of $C_{k^\star}$ contributed by points from $A_k$, $k \neq k^\star$).

## 5   Experiments 1: Controlled Setting

In our first set of experiments, we evaluate our approach, **R-LOCO**, by comparing it with three baseline methods: two local attribution techniques, **L-SV** and **LIME**, and **LOCO**, which serves as the global variant of our approach. We conduct experiments in a controlled environment where the ground-truth explanations are known, using independent features to assess each method's performance without confounding bias. Our objective is to demonstrate that **R-LOCO**, which lies between the local and global regimes, offers greater local fidelity than state-of-the-art local attribution methods and provides a more nuanced understanding of model behavior compared to global approaches.

**R-LOCO Variants:** We explored several variants of **R-LOCO** in our experiments, each employing different clustering algorithms to capture local patterns.

- **R-LOCO**: Uses AffinityPropagation clustering on the feature importance space. It is the default strategy of our proposal.
- **R-LOCO**$_{\text{TC}}$: Uses optimal clusters based on the base model structure, expected to perform better than other variants due to privileged model information. We use it as a **control baseline** to isolate the effect of the clustering in our method.
- **R-LOCO**$_{\text{tree}}$: Utilizes a tree-based algorithm that clusters by minimizing the variance on the feature importance space within each cell.
- **R-LOCO**$_{\text{IC}}$: Performs clustering directly in the input space rather than the feature importance space to investigate the benefits of clustering in feature importance space.

More details of the parameters for all methods can be found in the appendix C.

### 5.1   Models Setup

To ensure a rigorous evaluation, we selected a model setup that inherently allows us to determine ground-truth local explanations. Specifically, we chose models that are locally additive, enabling us to use their inherent structure to evaluate the quality of local attribution. To avoid biases that might arise from feature dependencies, we ensured that all features were independent. For data generation, we created a set of $n$ samples denoted by $\{\boldsymbol{X}_i\}_{i=1}^n$, drawn from a distribution $\boldsymbol{X} \sim P_{\boldsymbol{X}} = \prod_{i=j}^p P_{X_j}$. Using a model $f$, we then constructed a dataset $\mathcal{D}_n = \{(\boldsymbol{X}_i, f(\boldsymbol{X}_i))\}_{i=1}^n$.

The local attribution methods **L-SV** and **LIME** require both the model $f$ and dataset $\mathcal{D}_n$ to generate explanations. In contrast, **R-LOCO** only required fixed input-output pairs, $\mathcal{D}_n$. This highlights a key advantage of **R-LOCO**: it does not need direct access to the model's prediction function. Instead, it only needs a fixed dataset with predictions, making it suitable for scenarios where model access is restricted or for being applied directly to the data rather than a specific model. We conducted our experiments using three different models:

**1st-order model:** We used a piecewise linear model where local feature importance can be interpreted similarly to classical linear models. Let $\boldsymbol{X} = (X_1, \dots, X_6)$, with each component being independently distributed as $X_i \sim \mathcal{U}[-1, 1]$ for $i \in \{1, \dots, 6\}$. The predictor function is defined as follows:

$$f(\boldsymbol{X}) = (X_1 + X_2) \cdot \mathbf{1}_{X_6 \leq 0} + (X_3 + X_4) \cdot \mathbf{1}_{X_6 > 0} \tag{4}$$

**2nd-order model:** This model incorporates non-linear terms to introduce complexity while still being locally additive. Again, using independent variables $\boldsymbol{X} = (X_1, \dots, X_{10})$, where $X_i \sim \mathcal{U}[-1, 1]$ for all $i$, the predictor function was:

$$f(\boldsymbol{X}) = (X_1 + X_2^2 + \sqrt{|X_3|}) \cdot \mathbf{1}_{X_{10} \leq 0} + (X_4 + X_5^2 + \sqrt{|X_6|}) \cdot \mathbf{1}_{X_{10} > 0}$$

**2nd-order interaction model:** We increased the model complexity by including interaction effects, based on the function used in [Bénard et al., 2021]. The predictor was:

$$f(\boldsymbol{X}) = 3\sqrt{3} \times X_1 X_2 \, \mathbb{1}_{X_3>0} + \sqrt{3} \times X_4 X_5 \mathbb{1}_{X_3\leq0} + 3 \times X_6 X_7 \, \mathbb{1}_{X_8>0} + X_9 X_{10} \, \mathbb{1}_{X_8\leq0}.$$

Table 1: Performance metrics for methods across datasets from 1st-order to 2nd-order-interactions. TP: True Positives, FP: False Positives, NI attr.: Mean (q0.1 – q0.95) of non-important features attributions.

| Method | 1st-order | | | 2nd-order | | | 2nd-order-interact. | | |
|---|---|---|---|---|---|---|---|---|---|
| | TP | FP | NI attr. | TP | FP | NI attr. | TP | FP | NI attr. |
| **R-LOCO** | 0.97 | 0.03 | 0.02 (0.01–0.04) | 0.86 | 0.14 | 0.02 (0.01–0.04) | 0.67 | 0.33 | 0.03 (0.02–0.07) |
| **R-LOCO$_{TC}$** | 1.00 | 0.00 | 0.03 (0.03–0.03) | 1.00 | 0.00 | 0.03 (0.03–0.03) | 0.94 | 0.06 | 0.04 (0.03–0.04) |
| **R-LOCO$_{tree}$** | 0.90 | 0.10 | 0.05 (0.02–0.11) | 0.73 | 0.27 | 0.04 (0.02–0.06) | 0.50 | 0.50 | 0.06 (0.05–0.07) |
| **R-LOCO$_{IC}$** | 0.89 | 0.11 | 0.05 (0.02–0.11) | 0.74 | 0.26 | 0.04 (0.03–0.07) | 0.65 | 0.35 | 0.05 (0.04–0.07) |
| **LOCO** | 0.50 | 0.50 | 0.09 (0.09–0.09) | 0.50 | 0.50 | 0.05 (0.05–0.05) | 0.50 | 0.50 | 0.06 (0.05–0.07) |
| **L-SV** | 0.41 | 0.59 | 0.12 (0.08–0.18) | 0.30 | 0.70 | 0.08 (0.07–0.10) | 0.44 | 0.56 | 0.06 (0.05–0.09) |
| **LIME** | 0.53 | 0.47 | 0.12 (0.07–0.18) | 0.43 | 0.57 | 0.07 (0.05–0.10) | 0.44 | 0.56 | 0.06 (0.04–0.09) |

Table 2: Performance metrics for **R-LOCO** using different clustering algorithms across datasets from 1st-order to 2nd-order-interactions. TP: True Positives, FP: False Positives, NI attr.: Mean (q0.1 – q0.95) of non-important features attributions.

| Method | 1st-order | | | 2nd-order | | | 2nd-order-interact. | | |
|---|---|---|---|---|---|---|---|---|---|
| | TP | FP | NI attr. | TP | FP | NI attr. | TP | FP | NI attr. |
| **R-LOCO$_{KMeans-2}$** | 0.98 | 0.02 | 0.03 (0.03–0.03) | 0.92 | 0.08 | 0.04 (0.03–0.06) | 0.57 | 0.43 | 0.05 (0.03–0.09) |
| **R-LOCO$_{KMeans-4}$** | 0.97 | 0.03 | 0.03 (0.02–0.08) | 0.83 | 0.17 | 0.03 (0.03–0.06) | 0.62 | 0.38 | 0.04 (0.02–0.08) |
| **R-LOCO$_{KMeans-8}$** | 0.96 | 0.04 | 0.03 (0.02–0.06) | 0.93 | 0.07 | 0.03 (0.02–0.04) | 0.59 | 0.41 | 0.04 (0.02–0.08) |
| **R-LOCO$_{KMeans-20}$** | 0.97 | 0.03 | 0.02 (0.02–0.04) | 0.96 | 0.04 | 0.03 (0.02–0.04) | 0.64 | 0.36 | 0.04 (0.02–0.08) |
| **R-LOCO$_{Affinity-0.5}$** | 0.97 | 0.03 | 0.02 (0.01–0.04) | 0.86 | 0.14 | 0.02 (0.01–0.04) | 0.68 | 0.32 | 0.03 (0.02–0.07) |
| **R-LOCO$_{Affinity-0.6}$** | 0.97 | 0.03 | 0.02 (0.01–0.04) | 0.86 | 0.14 | 0.02 (0.01–0.04) | 0.67 | 0.33 | 0.03 (0.02–0.07) |
| **R-LOCO$_{Affinity-0.9}$** | 0.97 | 0.03 | 0.02 (0.01–0.04) | 0.86 | 0.14 | 0.02 (0.01–0.04) | 0.67 | 0.33 | 0.03 (0.02–0.07) |

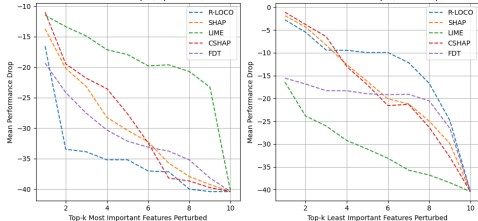

Figure 1: *Diabetes* dataset.

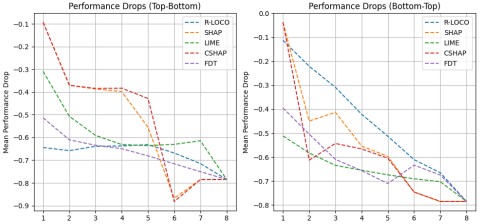

Figure 2: *California* dataset.

## 5.2 Evaluation

For each model, we identify the $k$ variables that contribute locally to the prediction. We exclude the variables used for defining the region (e.g., in the 1st order model, we discard the variable $X_6$), as these variables may also be interpreted as globally important, which could bias the analysis. Instead, we focus solely on the $k$ variables directly involved in the computation of the final prediction. As a measure of true importance, we use the sum of the absolute feature–coefficient products across all functional components that include the feature, analogous to classical functional ANOVA contributions. In some settings, we introduce dummy variables that are irrelevant to the model and mutually independent, included solely to assess how different methods assign importance to uninformative features.

To ensure a consistent comparison focused purely on importance magnitude, we normalise all attribution values. We divide the absolute value of each score by the total absolute sum per observation, thereby restricting all scores to the range $[0, 1]$ and disregarding their original directionality.

We then evaluate the performance of each method by determining how many of the $k$ truly local important variables (those with non-zero coefficients) are correctly identified by their top $k$ most

important features (True Positives) and how many irrelevant variables are incorrectly identified as important (False Positives). We also compute the average, as well as the 10th (q0.1) and 95th (q0.95) percentiles of the attributions assigned to non-important variables, which we denote as NI attributions. This assess the tendency of each method to over-attribute importance to irrelevant features.

## 5.3 Analysis

Table 1 presents the aggregated performance metrics of our experiments from 50 runs on the first three synthetic datasets. The key observations from these results are as follows: **Poor Performance of L-SV and LIME: L-SV** and **LIME** consistently perform poorly across all experiments, with true positive (TP) rates of 30% to 66% and false positive (FP) rates of 34% to 70% on Table 1. For example, in the 1st-order model, **L-SV** achieves a TP of 41% and FP of 59%, while **LIME** achieves TP of 53% and FP of 47%, indicating frequent misattributions. Moreover, their performance is almost identical to that of the global approach **LOCO**, which is not designed to highlight local importance.

**Superior Performance of R-LOCO: R-LOCO** outperforms the baselines, with significantly higher accuracy. In the 1st-order model, **R-LOCO** achieves a TP of 97% and FP of 3%. Although TP rates decrease slightly as model complexity grows, **R-LOCO** consistently outperforms the baselines.

**R-LOCO** assigns minimal importance to non-important features (1% to 4%), whereas **L-SV** and **LIME** assign 7% to 18%. This indicates **R-LOCO** more effectively distinguishes between important and non-important features.

**Impact of Clustering:** When using perfect clusters (**R-LOCO**$_{TC}$), **R-LOCO** achieves near-perfect performance, suggesting that clustering quality is the key factor for **R-LOCO**' success. Results of **R-LOCO**$_{IC}$ also show that clustering in the feature importance space (**R-LOCO**) yields better results compared to clustering in input space.

**Summary:** Overall, the experimental results show that the baseline methods **L-SV** and **LIME** consistently perform poorly to identify locally important features and have performance close to the global approach **LOCO**. In contrast, **R-LOCO** and its variants significantly outperform these traditional local attribution methods, providing more locally faithful feature attributions. The effectiveness of **R-LOCO** largely depends on the quality of the clustering process, particularly when clustering is performed in the feature importance space.

## 5.4 Impact of Clustering in R-LOCO

To illustrate how clustering quality can be assessed in our method, we present a qualitative example using the first-order model. We focus on an observation where $X_6$ is positive, implying that the only locally important variables are $X_3$ and $X_4$.

Figure 3 compares feature attributions from **R-LOCO** and **R-LOCO**$_{IC}$ (which clusters in the input space), highlighting the variance of feature-importance scores within each cluster. Both methods identify $X_3$ and $X_4$ as the most important variables, but **R-LOCO** exhibits substantially lower intra-cluster variance. Specifically, the average variance across all observations and features is $0.002$ for **R-LOCO**, compared to $0.012$ for **R-LOCO**$_{IC}$. This lower variance suggests more consistent attributions

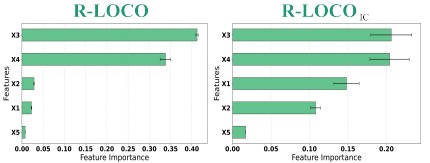

Figure 3: Feature attributions from **R-LOCO** and **R-LOCO**$_{IC}$ on the first-order model, highlighting the intra-group variance of feature importance.

within clusters, indicating higher clustering quality. By assessing the intra-cluster variance, practitioners can evaluate the clustering quality and make informed choices about clustering methods.

To further assess interpretability, we train a decision tree to predict the cluster assignments. This yields a transparent and human-readable proxy of the regions identified by each method. Interestingly, **R-LOCO** discovers regions aligned with the true data-generating process by first splitting on the sign of $X_6$, whereas clustering in the input space fails to do so, prioritising other non-important variables. The learned tree representation is displayed in §F.

**Parameter Variations:** To further assess clustering robustness, we conducted additional experiments using KMeans with cluster counts of 2, 4, 8, and 20, and Affinity Propagation with damping values of 0.5, 0.6, and 0.9.

Table 2 summarizes these findings, underscoring the robustness of **R-LOCO** to clustering variations.

Our key observations are:

- For both first and second-order models, as long as intra-cluster variance was minimized and clusters contained enough points, performance remained stable.
- Increasing the number of clusters had little adverse effect, provided cluster sizes were sufficiently large.
- In second-order interaction models with four ground-truth clusters, performance only declined when clusters were smaller than the true regions. Assigning more clusters than ground-truth also had minimal impact.

These results are consistent with Theorem 4.1, which states that local accuracy may be achieved once the identified cluster is part of the ground-truth cluster or is only minimally contaminated.

### 5.4.1 Theoretical Analysis

To formalize the advantage of clustering in importance space, we analyze the first-order model (§5.1) with independent features $X_i \sim \mathcal{U}[-1, 1]$ for $i = 1, \ldots, 6$ and the function $f(\mathbf{X}) = (X_1 + X_2) \cdot \mathbf{1}_{X_6 \leq 0} + (X_3 + X_4) \cdot \mathbf{1}_{X_6 > 0}$. This function operates in two distinct regimes based on the "switching" variable $X_6$: $R_A := \{\mathbf{x} \mid X_6 \leq 0\}$ and $R_B := \{\mathbf{x} \mid X_6 > 0\}$. We compare the K-Means ($K = 2$) decision rules derived from clustering in the input space versus the R-LOCO importance space.

**Clustering in Input Space.** The true cluster centroids are the conditional means $\mathbb{E}[\mathbf{X} \mid R_A]$ and $\mathbb{E}[\mathbf{X} \mid R_B]$, defined as $C_A = (0, 0, 0, 0, 0, -0.5)$ and $C_B = (0, 0, 0, 0, 0, 0.5)$. This follows from feature independence ($\mathbb{E}[X_i \mid R_A] = \mathbb{E}[X_i] = 0$ for $i \leq 5$) and the uniform distribution ($\mathbb{E}[X_6 \mid X_6 \leq 0] = -0.5$). The K-Means separating direction is $C_A - C_B = (0, 0, 0, 0, 0, -1)$. The decision rule, $\mathbf{x} \in R_A$ if $d(\mathbf{x}, C_A)^2 < d(\mathbf{x}, C_B)^2$, simplifies to a condition based *only* on $X_6$. Therefore, input-space clustering merely rediscovers the "switching" variable $X_6$. The variability from the functionally active features $(X_1, \ldots, X_4)$ acts as noise, hindering a clean separation.

**Clustering in R-LOCO Importance Space.** Each sample $\mathbf{X}$ is mapped to its R-LOCO importance vector $\gamma(\mathbf{X}) = (\Delta_1, \ldots, \Delta_6)$, where $\Delta_j(\mathbf{X}) = (f(\mathbf{X}) - f_{-j}(\mathbf{X}))^2$ and $f_{-j}(\mathbf{X}) = \mathbb{E}[f(\mathbf{X}) \mid \mathbf{X}_{-j}]$. The importance-space centroids for $R_A$ and $R_B$ are $C_{\gamma,A} = \mathbb{E}[\gamma(\mathbf{X}) \mid R_A] = \left(\frac{1}{3}, \frac{1}{3}, 0, 0, 0, \frac{1}{3}\right)$ and $C_{\gamma,B} = \mathbb{E}[\gamma(\mathbf{X}) \mid R_B] = \left(0, 0, \frac{1}{3}, \frac{1}{3}, 0, \frac{1}{3}\right)$, respectively. See G for detailed derivations. The K-Means separating direction in this space is: $C_{\gamma,A} - C_{\gamma,B} = \left(\frac{1}{3}, \frac{1}{3}, -\frac{1}{3}, -\frac{1}{3}, 0, 0\right)$. This decision rule *only* depends on the components $(\Delta_1, \Delta_2, \Delta_3, \Delta_4)$ corresponding to the locally active features. Clustering in importance space successfully identifies **functional similarity**:

- $R_A$ samples have large $(\Delta_1, \Delta_2)$ and $\Delta_{3,4} \approx 0$.
- $R_B$ samples have large $(\Delta_3, \Delta_4)$ and $\Delta_{1,2} \approx 0$.

In short, input-space clustering is confounded by noise from irrelevant features, while importance-space clustering is robust to this noise and correctly groups samples by their underlying explanatory behaviour.

**Remark: Importance Representation and Failure Cases** We note however a counter-example: $f(X_1, X_2) = X_2 \cdot \text{sgn}(X_1)$, with regimes $R_A = \{X_1 < 0\}$ and $R_B = \{X_1 \geq 0\}$. If we use the squared-loss importance $\Delta_j = (f(\mathbf{X}) - f_{-j}(\mathbf{X}))^2$, we find $f_{-1} = f_{-2} = 0$. This results in an importance vector $\gamma(\mathbf{X}) = (\Delta_1, \Delta_2) = (f(\mathbf{X})^2, f(\mathbf{X})^2) = (X_2^2, X_2^2)$. This representation is identical for both regimes, making them inseparable. The issue is that the squared loss $\Delta_j$ is **sign-agnostic** and discards the directional information that distinguishes the regimes.

This motivates enriching the importance representation. For instance, including the signed R-LOCO residual, $\delta_j(\mathbf{X}) = f(\mathbf{X}) - f_{-j}(\mathbf{X})$, yields an enriched vector:

$$\gamma'(\mathbf{X}) = (\Delta_1, \Delta_2, \delta_1, \delta_2) = (X_2^2, X_2^2, f(\mathbf{X}), f(\mathbf{X})).$$

This enriched vector maps the two regimes to distinct, separable locations:
$$\gamma'(\mathbf{X}) = (X_2^2, X_2^2, -X_2, -X_2) \text{ if } X_1 < 0, \text{ and } (X_2^2, X_2^2, X_2, X_2) \text{ if } X_1 \geq 0.$$

In this space, the regimes are easily distinguished by a clustering algorithm, underscoring the importance of the chosen representation.

## 6    Experiments 2: Real-world Settings

In the preceding sections, we focused on the piecewise model setting, providing both theoretical and empirical evidence that **L-SV** and **LIME** can fail under ideal conditions (e.g., exact computation and independence assumptions). We also introduced **R-LOCO** to address these shortcomings in a controlled context.

In this section, we extend our analysis to real-world tabular datasets, where **feature dependencies naturally arise**, and use widely adopted, high-performing models, XGBoost, for tabular data. We introduce two additional baselines:

- **FDT Laberge et al. [2024]:** This approach is conceptually related to ours in that it seeks "regional" explanations; however, its objective is to find regions where multiple explanation methods agree. In contrast, **R-LOCO** aims to identify zones of influence that are distinct to the model's behavior, using predictive loss as the criterion.
- **Clustering SHAP Values (CSHAP):** As described in the original SHAP paper [Lundberg et al., 2018], **L-SV** can be clustered to identify distinct zones of influence. Since SHAP does not directly employ group-wise averages, we adapt it for comparison by clustering SHAP values or **L-SV** (using the same algorithm as R-LOCO) and then computing averages for each group.

To evaluate the faithfulness of these methods, we measure the change in predictive performance when *zero-masking* features deemed most important (top-$k$) and least important (bottom-$k$) by each approach. See §H for details. A larger performance drop when masking the most important features indicates higher fidelity, whereas a smaller performance drop when masking the least important features is desirable.

Figures 1 and 2 show the results on the *Diabetes* and *California* datasets from Dua and Graff [2017], respectively. In Figure 1, our method consistently outperformed all baselines. While FDT performed best among the baselines, clustering SHAP values (CSHAP) did not outperform standard SHAP. This highlights that, unlike LOCO features, SHAP values lack sufficient information to effectively identify the distinct zone of influence of the given model, and underscores the critical importance of selecting appropriate inputs for clustering. In Figure 2, our method was superior for identifying and perturbing the top three most important features, and FDT was slightly better for the remaining, but **R-LOCO** outperformed when removing non-important features, indicating superior region definition. Finally, as shown in Appendix I, our method is also significantly faster at inference, since it only requires assigning a cluster to compute feature importance.

## 7    Conclusion

We presented a rigorous evaluation of widely used local attribution methods in settings where the true local features are known. Our findings reveal that, despite their intuitive appeal, common local methods (**L-SV**, **LIME**) can fail even under ideal conditions with no feature dependencies and perfect estimations. To address this, we proposed a *regional* method that partitions the input space into homogeneous regions—thus enabling a hybrid approach combining the fidelity of global LOCO importance with more localized attributions. Unlike existing baselines, our framework offers a clear interpretation of a feature's contribution as the increase in error when that feature is removed.

Our method lies between fully global and purely local approaches: avoiding the instability of pointwise explanations while still preserving instance-level insights, which purely global approaches often lack. **R-LOCO** implicitly assumes each region of the model can be approximated by an additive structure. A key challenge is selecting and refining these regions. One promising direction is to enrich the feature-importance representation (used for clustering) with second- or higher-order interactions, thus capturing more complex dependencies. While our initial analysis focused on independent variables for clarity, we have demonstrated that the approach also outperforms baselines on real-world datasets where dependencies naturally arise.

## Acknowledgements

We thank the anonymous reviewers for their valuable suggestions, which helped improve this work, especially regarding the counterexample illustrating the failure of the base feature-importance mapping.

## Disclaimer

This work was in part conducted during Salim I. Amoukou's PhD at LaMME, Université Paris-Saclay, in collaboration with the Stellantis Group.

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

# A Proof of Theorem 2.1

**Theorem A.1.** *Let $f$ be a piecewise linear function with $m$ components defined by the collection $\{f_{|A_1}, \ldots, f_{|A_k}\}$, where $\cup_{k=1}^m A_k = \mathcal{X}$. The regions $A_k$ are disjoint hyperrectangles, specifically $A_k = \bigotimes_{i=1}^p A_{i,k}$, where $A_{i,k} = [l_{i,k}, r_{i,k}]$ with $l_{i,k}, r_{i,k} \in \overline{\mathbb{R}}$. Each component $f_{|A_k}$ is represented as $f_k(\boldsymbol{X}) = \sum_{i=1}^p a_{i,k} X_i + b_k$, where the coefficients $a_{i,k}$ and $b_k$ are real numbers. Consequently, $f$ is defined as:*

$$f(\boldsymbol{X}) = \sum_{k=1}^m \left( \sum_{i=1}^p a_{i,k} X_i + b_k \right) \mathbb{1}_{A_k}(\boldsymbol{X}).$$

*Consider an observation $\mathbf{x} = (x_1, \ldots, x_p)$ located in a specific region $A_{k^\star}$ for some $k^\star \in \{1, \ldots, m\}$. The model's prediction is therefore determined solely by the local model $f_{k^\star}$, i.e., $f(\mathbf{x}) = f_{k^\star}(\mathbf{x})$, and the set of truly active features is $J_{k^\star} = \{i \in [p] \mid a_{i,k^\star} \neq 0\}$. We also define the set of features important globally $G = \{i \in [p] \mid \exists k \text{ s.t. } A_{i,k} \neq \mathbb{R}\}$. Let the covariates $X_i$ be mutually independent. The Local Shapley Value (SV) for the feature-value $X_l = x_l$ is equal to*

$$\phi_{x_l} = \sum_{k=1}^m \phi_{x_l}^k,$$

*and $\phi_{x_l}^k$ is defined as*

$$\phi_{x_l}^k = \left( \frac{\mathbb{1}_{A_{l,k}}(x_l)}{\mathbb{P}(X_l \in A_{l,k})} - 1 \right) \sum_{S \subseteq D \setminus \{l\}} w(S) v_k(S)$$

$$+ a_{l,k} \left( x_l - \frac{\mathbb{E}\left[ X_l \mathbb{1}_{A_{l,k}}(X_l) \right]}{\mathbb{P}(X_l \in A_{l,k})} \right) \sum_{S \subseteq D \setminus \{l\}} w(S) \times \prod_{i \in S \cup l} \mathbb{1}_{A_{i,k}}(x_i) \prod_{j \in \overline{S}} \mathbb{P}(X_i \in A_{i,k}), \quad (5)$$

*where $w(S) = \frac{1}{p} \binom{|D|-1}{|S|}^{-1}$ and $v_k(S) = \mathbb{E}[f_k(\boldsymbol{X}) \mathbb{1}_{A_k}(\boldsymbol{X}) \mid \boldsymbol{X}_S = \boldsymbol{x}_S]$. Equation (5) demonstrates that even if the model only uses $f_{k^\star}(\boldsymbol{x})$ for a given observation $\boldsymbol{x}$, the Local SV $\boldsymbol{x}$ may depend on the coefficients of the unused linear models $f_k$ for $k \in \{1, \ldots, m\} \setminus \{k^\star\}$, which are not globally important neither.*

*Proof.* Let's assume that $\prod_{i=1}^p \mathbb{P}(X_i \in A_{i,k}) > 0$ and intercepts $b_k = 0$ for all $k = 1, \ldots, m$ without loss of generality. Given an observation $\boldsymbol{x} = (x_1, \ldots, x_p)$, we consider the Shapley Value of a feature-value $X_l = x_l$ defined as

$$\phi_{\boldsymbol{x}_l} = \sum_{S \subseteq D \setminus \{l\}} w(S) \Big[ \Delta_l(S) \Big],$$

where $w(S) = \frac{1}{p} \binom{|D|-1}{|S|}^{-1}$, $\Delta_l(S) = v(S \cup l) - v(S)$ represent the marginal contribution, and $v(S) = \mathbb{E}\left[ f(\boldsymbol{X}) | \boldsymbol{X}_S = \boldsymbol{x}_S \right]$. Note that we can decompose $v(S)$ into $m$ separate terms as following:

$$v(S) = \mathbb{E} \left[ \sum_{k=1}^m \left( \sum_{i=1}^p a_{i,k} X_i \right) \mathbb{1}_{A_k}(\boldsymbol{X}) \mid \boldsymbol{X}_S = \boldsymbol{x}_S \right]$$

$$= \sum_{k=1}^m \mathbb{E} \left[ \left( \sum_{i=1}^p a_{i,k} X_i \right) \mathbb{1}_{A_k}(\boldsymbol{X}) \mid \boldsymbol{X}_S = \boldsymbol{x}_S \right]$$

$$= \sum_{k=1}^m v_k(S)$$

Hence, we can decompose the Shapley Value of $\phi_{\boldsymbol{x}_l}$ due to the linearity property of SV as

$$\phi_{x_l} = \sum_{k=1}^m \phi_{x_l}^k,$$

where $\phi_{x_l}^k$ corresponds to the SV compute using the value function $v_k(S)$. Therefore, we only need to prove that $\phi_{x_l}^k$ for $k \neq k^\star$ is not necessarily null to prove the Theorem. We have

$$
\begin{aligned}
v_k(S) &= \mathbb{E}\left[ \left( \sum_{i=1}^p a_{i,k} X_i \right) \mathbb{1}_{A_k}(\boldsymbol{X}) \mid \boldsymbol{X}_S = \boldsymbol{x}_S \right] \\
&= \mathbb{E}\left[ \left( \sum_{i \in S} a_{i,k} X_i + \sum_{i \in \bar{S}} a_{i,k} X_i \right) \prod_{j=1}^p \mathbb{1}_{A_{j,k}}(X_j) \mid \boldsymbol{X}_S = \boldsymbol{x}_S \right] \\
&= \left( \sum_{i \in S} a_{i,k} \boldsymbol{x}_i \right) \prod_{j \in S} \mathbb{1}_{A_{j,k}}(x_j) \prod_{j \in \bar{S}} \mathbb{P}(X_j \in A_{j,k}) \\
&\quad + \sum_{i \in \bar{S}} a_{i,k} \mathbb{E}\left[ X_i \mathbb{1}_{A_{i,k}}(X_i) \right] \prod_{j \in \bar{S}:j \neq i} \mathbb{P}(X_j \in A_{j,k}) \prod_{j \in S} \mathbb{1}_{A_{j,k}}(x_j).
\end{aligned}
$$

Similarly, we can write $v(S \cup l)$ as:

$$
\begin{aligned}
v_k(S \cup l) &= \left( \sum_{i \in S \cup l} a_{i,k} x_i \right) \prod_{j \in S \cup l} \mathbb{1}_{A_{j,k}}(x_j) \prod_{j \in \overline{S \cup l}} \mathbb{P}(X_j \in A_{j,k}) \\
&\quad + \sum_{i \in \overline{S \cup l}} a_{i,k} \mathbb{E}\left[ X_i \mathbb{1}_{A_{i,k}}(X_i) \right] \prod_{j \in \overline{S \cup l}:j \neq i} \mathbb{P}(X_j \in A_{j,k}) \prod_{j \in S \cup l} \mathbb{1}_{A_{j,k}}(x_j) \\
&= \left( \sum_{i \in S} a_{i,k} x_i \right) \prod_{j \in S} \mathbb{1}_{A_{j,k}}(x_j) \prod_{j \in \overline{S}} \mathbb{P}(X_j \in A_{j,k}) \times \frac{\mathbb{1}_{A_{l,k}}(x_l)}{\mathbb{P}(X_l \in A_{l,k})} \\
&\quad {\color{teal} + a_{l,k} x_l \prod_{j \in S} \mathbb{1}_{A_{j,k}}(x_j) \prod_{j \in \overline{S}} \mathbb{P}(X_j \in A_{j,k}) \times \frac{\mathbb{1}_{A_{l,k}}(x_l)}{\mathbb{P}(X_l \in A_{l,k})}} \\
&\quad + \sum_{i \in \overline{S}} a_{i,k} \mathbb{E}\left[ X_i \mathbb{1}_{A_{i,k}}(X_i) \right] \prod_{j \in \overline{S}:j \neq i} \mathbb{P}(X_j \in A_{j,k}) \prod_{j \in S} \mathbb{1}_{A_{j,k}}(x_j) \times \frac{\mathbb{1}_{A_{l,k}}(x_l)}{\mathbb{P}(X_l \in A_{l,k})} \\
&\quad {\color{red} - a_{l,k} \mathbb{E}\left[ X_l \mathbb{1}_{A_{l,k}}(X_l) \right] \prod_{j \in \overline{S}:j \neq l} \mathbb{P}(X_j \in A_{j,k}) \prod_{j \in S} \mathbb{1}_{A_{j,k}}(x_j) \times \frac{\mathbb{1}_{A_{l,k}}(x_l)}{\mathbb{P}(X_l \in A_{l,k})}}
\end{aligned}
$$

The terms highlighted in red and teal respectively represent the negative and positive contributions of the variable $X_l = x_l$ in $v_k(S \cup l)$, the other terms will be put together to form $v_k(S)$ as follows

$$
\begin{aligned}
v_k(S \cup l) &= \frac{\mathbb{1}_{A_{l,k}}(x_l)}{\mathbb{P}(X_l \in A_{l,k})} v_k(S) \\
&\quad + \frac{1}{\mathbb{P}(X_l \in A_{l,k})} \prod_{j \in S \cup l} \mathbb{1}_{A_{j,k}}(x_j) \prod_{j \in \overline{S}} \mathbb{P}(X_j \in A_{j,k}) \times a_{l,k} \left( x_l - \frac{\mathbb{E}\left[ X_l \mathbb{1}_{A_{l,k}}(X_l) \right]}{\mathbb{P}(X_l \in A_{l,k})} \right).
\end{aligned}
$$

Hence, the marginal contribution of coalition $S$ of the SV $\phi_{x_l}^k$ is equal to:

$$
\begin{aligned}
\Delta_l^k(S) &= v_k(S \cup l) - v_k(S) \\
&= v_k(S) \left( \frac{\mathbb{1}_{A_{l,k}}(x_l)}{\mathbb{P}(X_l \in A_{l,k})} - 1 \right) \\
&\quad + \prod_{j \in S \cup l} \mathbb{1}_{A_{j,k}}(x_j) \prod_{j \in \overline{S}} \mathbb{P}(X_j \in A_{j,k}) \times a_{l,k} \left( x_l - \frac{\mathbb{E}\left[ X_l \mathbb{1}_{A_{l,k}}(X_l) \right]}{\mathbb{P}(X_l \in A_{l,k})} \right)
\end{aligned}
$$

Finally, we have

$$\phi_{x_l}^k = \left( \frac{\mathbb{1}_{A_{l,k}}(x_l)}{\mathbb{P}(X_l \in A_{l,k})} - 1 \right) \sum_{S \subseteq D \setminus \{l\}} w(S) v_k(S)$$

$$+ a_{l,k} \left( x_l - \frac{\mathbb{E}\left[ X_l \mathbb{1}_{A_{l,k}}(X_l) \right]}{\mathbb{P}(X_l \in A_{l,k})} \right) \sum_{S \subseteq D \setminus \{l\}} w(S) \times \prod_{j \in S \cup l} \mathbb{1}_{A_{j,k}}(x_j) \prod_{j \in \overline{S}} \mathbb{P}(X_j \in A_{j,k})$$

$\square$

## B Proof of Theorem 4.1

**Theorem B.1.** *Let $f$ be the piecewise linear function from Theorem 2.1. When the **R-LOCO** clustering algorithm identifies clusters $C_{k^\star}$ within regions $A_{k^\star}$, the R-LOCO attributions for $x = (x_1, \ldots, x_p) \in A_{k^\star}$, with $k^\star \in \{1, \ldots, m\}$ (the active region), are solely influenced by $f_{k^\star}$. They do not depend on coefficients any $a_{i,k'}$ from other regions $A_{k'}$, where $k' \neq k^\star$.*

*Proof.* The R-LOCO attribution for a feature $X_j$ within the region defined by cluster $C_{k^\star}$ is given by the average change in the conformity score $V$ when that feature is removed:

$$\hat{\Psi}_j(\hat{P}_{C_{k^\star}}) = \frac{1}{|C_{k^\star}|} \sum_{(\mathbf{x}_l, y_l) \in C_{k^\star}} [V(f(\mathbf{x}_l), y_l) - V(f_{-j}(\mathbf{x}_l), y_l)]$$

where $f$ is the full model and $f_{-j}(\mathbf{x}) = \mathbb{E}[f(\mathbf{X})|\mathbf{X}_{-j} = \mathbf{x}_{-j}]$ is the model with feature $j$ marginalized out.

The theorem's central assumption is that the clustering algorithm is that the identified cluster $C_{k^\star}$ is a subset of the true model region $A_{k^\star}$, i.e., $C_{k^\star} \subseteq A_{k^\star}$. We analyze the implications of this condition on both terms in the R-LOCO formula.

1. **Analysis of the full model, $f(\mathbf{x}_l)$:** For any observation $(\mathbf{x}_l, y_l) \in C_{k^\star}$, our core assumption implies $\mathbf{x}_l \in A_{k^\star}$. By definition, the global piecewise model $f(\mathbf{x}) = \sum_{k=1}^{m} f_k(\mathbf{x}) \mathbb{1}_{\mathbf{x} \in A_k}$ simplifies to the active local model for any point within its region. Therefore, for all $\mathbf{x}_l \in C_{k^\star}$:
   $$f(\mathbf{x}_l) = f_{k^\star}(\mathbf{x}_l) = \mathbf{a}_{k^\star}^T \mathbf{x}_l + b_{k^\star}$$
   This shows that the first term, $V(f(\mathbf{x}_l), y_l)$, depends exclusively on the parameters of the local model $f_{k^\star}$.

2. **Analysis of the reduced model, $f_{-j}(\mathbf{x}_l)$:** The proof is particularly concerned with features $j$ that are not globally important, meaning they are not used to define the region boundaries. Let $G = \{i \in [p] \mid \exists k \text{ s.t. } A_{i,k} \neq \mathbb{R}\}$ be the set of globally important features. If $j \notin G$, the condition $\mathbf{x} \in A_{k^\star}$ does not depend on the value of $x_j$.

   Consequently, for any $\mathbf{x}_l \in C_{k^\star} \subseteq A_{k^\star}$, any other point $\mathbf{X}$ that agrees with $\mathbf{x}_l$ on all other coordinates (i.e., $\mathbf{X}_{-j} = \mathbf{x}_{l,-j}$) must also lie within $A_{k^\star}$. This implies:
   $$f_{-j}(\mathbf{x}_l) = \mathbb{E}[f(\mathbf{X})|\mathbf{X}_{-j} = \mathbf{x}_{l,-j}]$$
   $$= \mathbb{E}\left[ \sum_{k=1}^{m} f_k(\mathbf{X}) \mathbb{1}_{\mathbf{X} \in A_k} \Big| \mathbf{X}_{-j} = \mathbf{x}_{l,-j} \right]$$
   $$= \mathbb{E}[f_{k^\star}(\mathbf{X})|\mathbf{X}_{-j} = \mathbf{x}_{l,-j}]$$
   The last step holds because the indicator $\mathbb{1}_{\mathbf{X} \in A_{k^\star}}$ is always 1 under the conditioning, while all other indicators $\mathbb{1}_{\mathbf{X} \in A_{k'}}$ for $k' \neq k^\star$ are 0. The resulting expression for $f_{-j}(\mathbf{x}_l)$ depends only on the definition of $f_{k^\star}$.

Since both $f(\mathbf{x}_l)$ and $f_{-j}(\mathbf{x}_l)$ depend solely on the parameters $(\mathbf{a}_{k^\star}, b_{k^\star})$ for any observation in $C_{k^\star}$, each term in the summation for $\hat{\Psi}_j$ is a function of $f_{k^\star}$ alone. The attribution, being an average of these terms, is therefore completely independent of the coefficients $(\mathbf{a}_{k'}, b_{k'})$ from any other region $k' \neq k^\star$. This concludes the proof.

## B.1 Analysis under Cluster Contamination.

We now consider the practical scenario where the cluster is contaminated, meaning $C_{k^\star} \not\subseteq A_{k^\star}$. Let $C'_{k^\star} = C_{k^\star} \cap A_{k^\star}$ be the set of correctly clustered points and $C''_{k^\star} = C_{k^\star} \setminus A_{k^\star}$ be the set of contaminating points. For any point $\mathbf{x}_l \in C''_{k^\star}$, there exists some region $A_{k'}$ with $k' \neq k^\star$ such that $\mathbf{x}_l \in A_{k'}$. For these contaminating points, the global model evaluates to $f(\mathbf{x}_l) = f_{k'}(\mathbf{x}_l)$. Consequently, the R-LOCO summation for $\hat{\Psi}_j$ becomes a mixture:

$$\hat{\Psi}_j = \frac{1}{|C_{k^\star}|} \left( \sum_{\mathbf{x}_l \in C'_{k^\star}} \Delta_j(\mathbf{x}_l) + \sum_{\mathbf{x}_l \in C''_{k^\star}} \Delta_j(\mathbf{x}_l) \right)$$

where the terms $\Delta_j$ for points in $C'_{k^\star}$ depend on $f_{k^\star}$ as shown above, but the terms for points in $C''_{k^\star}$ depend on the parameters of other local models $f_{k'}$. As a result, the attribution is no longer local. It becomes "contaminated" by the influence of coefficients from the regions where the misclustered points originate. The magnitude of this effect is proportional to the fraction of contaminating points $|C''_{k^\star}|/|C_{k^\star}|$. $\qquad\square$

## C   Experimental Details

Each dataset is divided into three parts: training, calibration, and test. Initially, we split the data into a training set (75%) and a test set (25%). The training set is then further split evenly into a new training and calibration set. The calibration data is used by each method as background for computing approximations.

For the baselines, we compute **L-SV** using the exact explainer from the SHAP library.

For **LIME**, we utilize the LIME package with default parameters, as described in the original paper [Ribeiro et al., 2016a].

For **R-LOCO**, we use XGBoost (default parameters) as the base model to approximate the leave-one-out functions $\widehat{f}_0$ and $\widehat{f}_{-j}$. We use Affinity Propagation with damping = 0.8. Further implementation details will be given in the code repository. We split our data into two sets: one for model fitting and the other for estimating attribution quantities.

## D   Limitations of LIME: Detailled

The main idea behind LIME [Ribeiro et al., 2016b] is to approximate a complex model using a simpler, more interpretable model, such as a linear model, in the vicinity of a given input instance. Given a model $f$, the local explanation $\xi(\boldsymbol{x}^\star)$ of an instance $\boldsymbol{x}^\star$ is an interpretable model $g \in G$, where $G$ is the set of linear models, such that

$$\xi(\boldsymbol{x}^\star) = \arg \min_{g \in G} \mathcal{L}(f, g, \pi_{\boldsymbol{x}^\star}^h) + \Omega(g), \tag{6}$$

where $\mathcal{L}(f, g, \pi_{\boldsymbol{x}^\star}^h)$ measure of how unfaithful $g$ is in approximating $f$ over $\pi_{\boldsymbol{x}^\star}^h$, a measure of locality around $x^\star$ with width $h$, and $\Omega(g)$ is a measure of complexity of the local model $g$. The loss $\mathcal{L}$ is defined as

$$\mathcal{L}(f, g, \pi_{\boldsymbol{x}^\star}^h) = \sum_{x' \sim P'} \left[ f(x') - g(x') \right]^2 \pi_{\boldsymbol{x}^\star}^h(\boldsymbol{x}').$$

In the original implementation [Ribeiro et al., 2016b], $\pi_{\boldsymbol{x}^\star}^h$ is a Gaussian kernel, and the sum is taken over samples $\boldsymbol{x}' \sim P'$ where $P' = \prod_i P_{X_i}$ is the marginal law of the features.

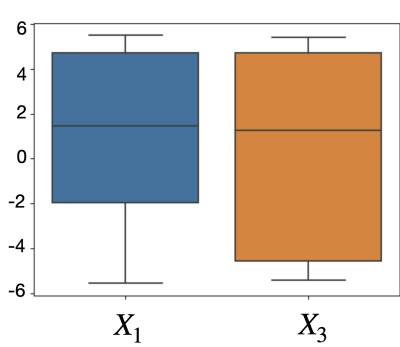

Figure 4: Distribution of LIME coefficients for $X_1$ and $X_3$ among observations with negative $x_5$ for the model (1)

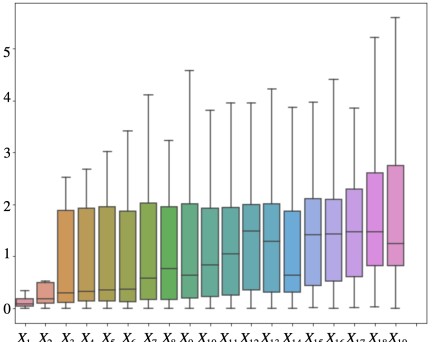

Figure 5: Variability of LIME coefficients wrt bandwidth selection on the German credit dataset ($n = 1000, p = 20$) from UCI [Dua and Graff, 2017]

A primary concern with LIME stems from its reliance on arbitrary heuristics in its definition. Specifically, choosing the sampling distribution $P'$ poses challenges, as the commonly used distribution disregards feature dependencies, and there is no guarantee that the model's local behavior on $P'$ will be consistent with that on the observed data. Another significant issue lies in defining the neighborhood $\pi_{\boldsymbol{x}^\star}^h$ and tuning the kernel width $h$, especially in high-dimension. The stability and sensitivity of LIME are heavily influenced by the selection of the perturbation sampling distribution, the definition of proximity $\pi_{\boldsymbol{x}^\star}^h$ and the bandwidth $h$, which may result in varying explanations for

the same instance under slightly varying settings. To illustrate this issue, we apply LIME on the piece-wise linear model defined in (1), with $a_1 = 0, a_2 = 2, a_3 = 0, a_4 = 5$.

In Figure 4, the distribution of LIME coefficients for $X_1$ and $X_3$ among observations with negative $x_6$ values shows that LIME assigns a non-null score to both $X_1$ and $X_3$, although the latter is not used locally for the displayed observations. Thus, LIME shares the same problem as L-SV in the piece-wise linear model with independent variables. It is also important to note that such discontinuities are not uncommon, as tabular data often contain discontinuities with categorical variables, and the meaning of constructing a linear approximation in such cases remains unclear. Besides this empirical evidence, we are currently working to compute the theoretical quantity of LIME coefficient for continuous piece-wise-linear functions, yielding impossible results similar to Thereom 2.1 for Local Shapley Values.

To demonstrate the sensitivity of LIME to bandwidth selection, we applied it to the German credit dataset ($n = 1000, p = 20$) from UCI [Dua and Graff, 2017]. We trained a RF on $80\%$ of the dataset and computed LIME coefficients on the remaining data using two bandwidths, $h$ and $h'$, of the proximity kernel $\pi_{\boldsymbol{x}^\star}^h$. We set $h$ using the median heuristic [Fukumizu et al., 2009, Flaxman et al., 2016, Garreau et al., 2017], where $h$ is the median of the pairwise distance of $||\boldsymbol{X}_i - \boldsymbol{X}_j||_2$ and assigned $h' = \frac{1}{2}h$. In Figure 5, we compare the relative absolute error of the LIME coefficients for all variables using the two bandwidths, i.e., $(\substack{h\\\boldsymbol{x}_i} - L_{\boldsymbol{x}_i}^{h'})/L_{\boldsymbol{x}_i}^h$, where $\substack{h\\\boldsymbol{x}_i}, \substack{h'\\\boldsymbol{x}_i}$ represent the LIME coefficient of the variable $X_i$ using bandwith $h, h'$ respectively. It reveals that the difference between the LIME coefficients could be much different after slightly modifying the bandwidth. Furthermore, we observed that $20\%$ of the coefficients also changed signs. In real-world scenarios, we lack information about the true local importance, making the bandwidth selection process indefinite. Moreover, LIME exhibits issues related to instability or irreproducibility. For example, Zhang et al. [2019], Zafar and Khan [2019], Visani et al. [2020] have shown that repeated runs of the same setting of the algorithm on the same model and data point can yield different results. This inconsistency stems from the randomness introduced during the generation of the synthetic sample around the input. Several works [Zafar and Khan, 2019, Zhou et al., 2021] attempt to address this issue, using asymptotic analysis of the method to identify the minimum number of the sampled observations required for stability. Another aspect of stability is related to input perturbations. Alvarez-Melis and Jaakkola [2018] show that nearby observations may have completely different LIME coefficients.

# E  Visual Description of R-LOCO

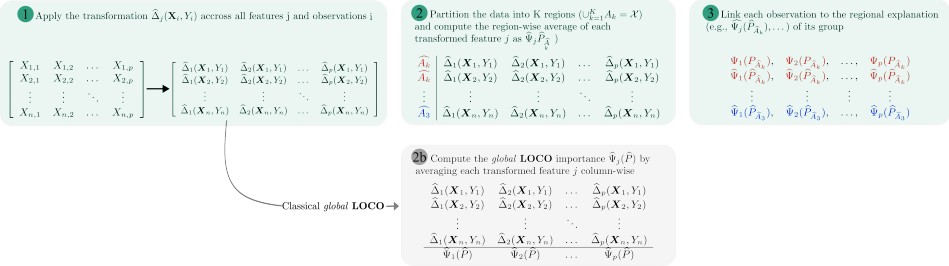

Figure 6: This diagram presents and contrasts our proposed method (**R-LOCO**) with the global approach (**LOCO**). Both approaches first transform the features into the feature importance space. The global approach (in *grey*) then averages the importance scores column-wise across all observations, while **R-LOCO** (in *green*) averages the scores within each cluster, providing a more localized attribution. However, note that nothing stops us from applying any other global methods after the clusters are identified. The estimation of the region-based attributions is very generic; we could simply learn a linear model within each identified region and use it as a global explanation, or apply any more advanced global attribution techniques. For example, we could re-estimate the $\widehat{f}_0$, and $\widehat{f}_{-j}$ of $\widehat{\Delta}_j(\boldsymbol{X}, Y) = V\left(\widehat{f}_0(\boldsymbol{X}), Y\right) - V\left(\widehat{f}_{-j}(\boldsymbol{X}), Y\right)$ used to estimate $\widehat{\Psi}_j(\widehat{P}_{\widehat{A}_k})$ within the corresponding cluster instead of using the one learned to identify the clusters. We found that both strategies perform similarly in our experiments.

# F    Learned Tree of Section of 5.4

The decision rules defining the regions shown in Section 5.4 are:

**Regions identified by R-LOCO in feature-importance space**

```
|--- X6 <= 0.00
|    |--- X5 <= -0.86
|    |    |--- class: 0
|    |--- X5 >  -0.86
|    |    |--- class: 0
|--- X6 >  0.00
|    |--- X4 <= -0.56
|    |    |--- class: 1
|    |--- X4 >  -0.56
|    |    |--- class: 0
```

**Regions identified by clustering in the input space**

```
|--- X5 <= -0.01
|    |--- X2 <= -0.00
|    |    |--- class: 0
|    |--- X2 >  -0.00
|    |    |--- class: 1
|--- X5 >  -0.01
|    |--- X6 <= 0.05
|    |    |--- class: 2
|    |--- X6 >  0.05
|    |    |--- class: 3
```

# G    Derivations of the centroids for the 1st-order model

To formalize the advantage of clustering in importance space, we analyze a model with independent features $X_i \sim \mathcal{U}[-1, 1]$ for $i = 1, \ldots, 6$ and the function:

$$f(\mathbf{X}) = (X_1 + X_2) \cdot \mathbf{1}_{X_6 \leq 0} + (X_3 + X_4) \cdot \mathbf{1}_{X_6 > 0}.$$

This function operates in two distinct regimes based on the "switching" variable $X_6$: $R_A \coloneqq \{\mathbf{x} \mid X_6 \leq 0\}$ and $R_B \coloneqq \{\mathbf{x} \mid X_6 > 0\}$. We compare the K-Means ($K = 2$) decision rules derived from clustering in the input space versus the R-LOCO importance space.

**Clustering in Input Space**   The true cluster centroids are the conditional means $\mathbb{E}[\mathbf{X} \mid R_A]$ and $\mathbb{E}[\mathbf{X} \mid R_B]$:

$$C_A = (0, 0, 0, 0, 0, -0.5), \quad C_B = (0, 0, 0, 0, 0, 0.5).$$

This follows from feature independence ($\mathbb{E}[X_i \mid R_A] = \mathbb{E}[X_i] = 0$ for $i \leq 5$) and the uniform distribution ($\mathbb{E}[X_6 \mid X_6 \leq 0] = -0.5$).

The K-Means separating direction is $C_A - C_B = (0, 0, 0, 0, 0, -1)$. The decision rule, $d(\mathbf{x}, C_A)^2 < d(\mathbf{x}, C_B)^2$, simplifies to a condition based *only* on $X_6$. Therefore, input-space clustering merely rediscovers the "switching" variable $X_6$. The variability from the functionally active features $(X_1, \ldots, X_4)$ acts as noise, hindering a clean separation.

**Clustering in R-LOCO Importance Space**   Each sample $\mathbf{X}$ is mapped to its R-LOCO importance vector $\gamma(\mathbf{X}) = (\Delta_1, \ldots, \Delta_6)$, where $\Delta_j(\mathbf{X}) = (f(\mathbf{X}) - f_{-j}(\mathbf{X}))^2$ and $f_{-j}(\mathbf{X}) = \mathbb{E}[f(\mathbf{Y}) \mid \mathbf{Y}_{-j} = \mathbf{X}_{-j}]$.

Consider $\mathbf{X} \in R_A$, where $X_6 \leq 0$ and $f(\mathbf{X}) = X_1 + X_2$. We find the components $\Delta_j$ and their expectations $\mathbb{E}[\Delta_j \mid R_A]$:

- $j = 1, 2$ **(Active):** $f_{-1}(\mathbf{X}) = \mathbb{E}[Y_1 + X_2] = X_2$.

- $\Delta_1 = (X_1 + X_2 - X_2)^2 = X_1^2$.
- $\mathbb{E}[\Delta_1 \mid R_A] = \mathbb{E}[X_1^2] = 1/3$. By symmetry, $\mathbb{E}[\Delta_2 \mid R_A] = 1/3$.

- $j = 3, 4, 5$ **(Inactive):** $f_{-3}(\mathbf{X}) = \mathbb{E}[X_1 + X_2] = X_1 + X_2$.
  - $\Delta_{3,4,5} = (X_1 + X_2 - (X_1 + X_2))^2 = 0$.
  - $\mathbb{E}[\Delta_{3,4,5} \mid R_A] = 0$.

- $j = 6$ **(Switching):** $f_{-6}(\mathbf{X}) = \mathbb{E}_{Y_6}[(X_1 + X_2)\mathbf{1}_{Y_6 \leq 0} + (X_3 + X_4)\mathbf{1}_{Y_6 > 0}] = \frac{1}{2}(X_1 + X_2) + \frac{1}{2}(X_3 + X_4)$.
  - $\Delta_6 = \left((X_1 + X_2) - \frac{1}{2}(X_1 + X_2 + X_3 + X_4)\right)^2 = \frac{1}{4}(X_1 + X_2 - X_3 - X_4)^2$.
  - $\mathbb{E}[\Delta_6 \mid R_A] = \frac{1}{4}\mathbb{E}[X_1^2 + X_2^2 + X_3^2 + X_4^2] = \frac{1}{4}(4 \cdot \frac{1}{3}) = 1/3$.

The importance-space centroid for $R_A$ is $C_{\gamma,A} = \left(\frac{1}{3}, \frac{1}{3}, 0, 0, 0, \frac{1}{3}\right)$. By symmetry, for $\mathbf{X} \in R_B$ (where $f(\mathbf{X}) = X_3 + X_4$), the centroid is $C_{\gamma,B} = \left(0, 0, \frac{1}{3}, \frac{1}{3}, 0, \frac{1}{3}\right)$.

**Decision Rule and Interpretation**    The K-Means separating direction in this space is:

$$C_{\gamma,A} - C_{\gamma,B} = \left(\frac{1}{3}, \frac{1}{3}, -\frac{1}{3}, -\frac{1}{3}, 0, 0\right).$$

This decision rule *only* depends on the components $(\Delta_1, \Delta_2, \Delta_3, \Delta_4)$ corresponding to the locally active features. Clustering in importance space successfully identifies **functional similarity**:

- $R_A$ samples have large $(\Delta_1, \Delta_2)$ and $\Delta_{3,4} \approx 0$.
- $R_B$ samples have large $(\Delta_3, \Delta_4)$ and $\Delta_{1,2} \approx 0$.

In short, input-space clustering is confounded by noise from irrelevant features and only finds the geometric switch $X_6$. Importance-space clustering is robust to this noise and correctly groups samples by their underlying explanatory behavior.

**Remark: Importance Representation and Failure Cases**    We note a known counter-example: $f(X_1, X_2) = X_2 \cdot \mathrm{sgn}(X_1)$, with regimes $R_A = \{X_1 < 0\}$ and $R_B = \{X_1 \geq 0\}$.

If we use the squared-loss importance $\Delta_j = (f(\mathbf{X}) - f_{-j}(\mathbf{X}))^2$, we find $f_{-1} = f_{-2} = 0$. This results in an importance vector $\gamma(\mathbf{X}) = (\Delta_1, \Delta_2) = (f(\mathbf{X})^2, f(\mathbf{X})^2) = (X_2^2, X_2^2)$. This representation is identical for both regimes, making them inseparable. The issue is that the squared loss $\Delta_j$ is **sign-agnostic** and discards the directional information that distinguishes the regimes.

This motivates enriching the importance representation. For instance, including the signed R-LOCO residual, $\delta_j(\mathbf{X}) = f(\mathbf{X}) - f_{-j}(\mathbf{X})$, yields an enriched vector:

$$\gamma'(\mathbf{X}) = (\Delta_1, \Delta_2, \delta_1, \delta_2) = (X_2^2, X_2^2, f(\mathbf{X}), f(\mathbf{X})).$$

This enriched vector maps the two regimes to distinct, separable locations:

$$\gamma'(\mathbf{X}) = \begin{cases} (X_2^2, X_2^2, -X_2, -X_2) & \text{if } X_1 < 0 \text{ (Regime } R_A), \\ (X_2^2, X_2^2, X_2, X_2) & \text{if } X_1 \geq 0 \text{ (Regime } R_B). \end{cases}$$

In this space, the regimes are easily distinguished by a clustering algorithm, underscoring the importance of the chosen representation.

# H    Evaluation methodology for Real-World Data

## H.1    Definitions

We begin by outlining the core components of our evaluation framework:

- **Dataset** $\mathcal{D}$: A collection of $n$ samples. Each sample is a pair $(\mathbf{x}, y)$, where $\mathbf{x}$ denotes the input and $y$ the ground-truth target.
- **Input** $\mathbf{x}$: A $d$-dimensional vector of numerical features.

- **Model** $f(\mathbf{x})$: The predictive model that maps $\mathbf{x}$ to a prediction.
- **Attribution Method** $A(\mathbf{x})$: A function that assigns a positive importance score to each of the $d$ input features. These scores correspond to normalized attributions from each explanation method (e.g., R-LOCO, SHAP).
- **Error Function** $E(\mathbf{x}, y)$: A metric quantifying the discrepancy between the model's prediction for $\mathbf{x}$ and the true label $y$:
  - **Classification:** $E(\mathbf{x}, y) = 1$ if the prediction is incorrect, 0 otherwise.
  - **Regression:** $E(\mathbf{x}, y)$ may be the absolute difference between the predicted and true values.

## H.2   Masking Procedure

To assess feature importance, we apply a *masking* (or *occlusion*) technique as follows:

1. For each input $\mathbf{x}$, compute the feature importance scores using $A(\mathbf{x})$.
2. Identify the indices of:
   - The top-$k$ most important features (**Top-$k$ Indices**).
   - The bottom-$k$ least important features (**Bottom-$k$ Indices**).
3. Create two modified versions of $\mathbf{x}$:

$$\mathbf{x}^{\text{masked,top-}k} : \text{Replace values at Top-}k \text{ indices with zero,}$$

$$\mathbf{x}^{\text{masked,bottom-}k} : \text{Replace values at Bottom-}k \text{ indices with zero.}$$

## H.3   Error Calculation

We compute the average model error over the dataset under three conditions:

**Original Average Error (Baseline):**

$$\text{AvgError}_{\text{Original}} = \frac{1}{n} \sum_{(\mathbf{x},y)\in\mathcal{D}} E(\mathbf{x}, y)$$

**Top-$k$ Masked Average Error:**

$$\text{AvgError}_{\text{Top-}k} = \frac{1}{n} \sum_{(\mathbf{x},y)\in\mathcal{D}} E(\mathbf{x}^{\text{masked,top-}k}, y)$$

**Bottom-$k$ Masked Average Error:**

$$\text{AvgError}_{\text{Bottom-}k} = \frac{1}{n} \sum_{(\mathbf{x},y)\in\mathcal{D}} E(\mathbf{x}^{\text{masked,bottom-}k}, y)$$

## H.4   Final Metrics for Figures

The values shown in Figures 1 and 2 represent the *change in model error* due to masking, computed as follows:

**Top-$k$ Error Change (Left Figure):**   Indicates the degradation in performance after removing the Top-$k$ most important features:

$$\text{ChangeInError}_{\text{Top}}(k) = \text{AvgError}_{\text{Original}} - \text{AvgError}_{\text{Top-}k}.$$

**Bottom-$k$ Error Change (Right Figure):**   Reflects the performance change after removing the Top-$k$ least important features:

$$\text{ChangeInError}_{\text{Bottom}}(k) = \text{AvgError}_{\text{Original}} - \text{AvgError}_{\text{Bottom-}k}.$$

# I Runtime and Memory Usage Analysis

We find that R-LOCO is significantly faster at inference time than the baselines. While our method incurs additional computational cost during training (due to clustering and model fitting), it performs inference with minimal overhead, as only the cluster assignment is required per sample. A similar pattern is observed for memory usage.

In terms of scalability, R-LOCO's **training** cost grows approximately linearly with the number of features. This behaviour is consistent with widely used explanation approaches such as SHAPLoss and its variants [Covert et al., 2020a]. Importantly, this training step is performed only once.

Moreover, R-LOCO can be further optimised for tree-based models using efficient approximation methods such as *Projected Forests* [Bénard et al., 2021, I. Amoukou and Brunel, 2022], where a single model can be reused to compute multiple conditional expectations efficiently.

| Method | Train (s) | Infer (s) | Total (s) | Train (MB) | Infer (MB) | Peak (MB) |
|--------|-----------|-----------|-----------|------------|------------|-----------|
| R-LOCO | 33.521 | 0.032 | 33.553 | 1687.9 | 8.6 | 2743.0 |
| LIME | 0.038 | 34.684 | 34.722 | 0.5 | 2.3 | 1.0 |
| SHAP | 0.002 | 20.698 | 20.700 | 0.0 | 0.8 | 6.8 |

Table 3: Runtime and memory usage comparison on the California dataset.

| Method | Train (s) | Infer (s) | Total (s) | Train (MB) | Infer (MB) | Peak (MB) |
|--------|-----------|-----------|-----------|------------|------------|-----------|
| R-LOCO | 1.611 | 0.010 | 1.621 | 12.2 | 0.5 | 1.7 |
| SHAP | 0.002 | 1.433 | 1.436 | 0.0 | 145.0 | 16.7 |
| LIME | 0.008 | 0.838 | 0.846 | 0.2 | 1.3 | 0.4 |

Table 4: Runtime and memory usage comparison on the Diabetes dataset.

