# OpenReview forum: "Regional Explanations: Bridging Local and Global Variable Importance"
_NeurIPS.cc/2025/Conference — NeurIPS 2025 poster_

### Official Review · Reviewer_v4Qd · 2025-06-09

**Clarity:** 3
**Significance:** 2
**Originality:** 2
**Rating:** 4
**Confidence:** 4

**Summary:**

The authors first discuss a major limitation of the vanilla-SHAP local explanation method : when the model is piece-wise linear over regions $A_1, A_2, \ldots$, the local importance of a feature in $A_k$ can be polluted by model behavior in other regions, leading to counterintuitive interpretations.

To address this limitation, it is proposed to first compute a point-wise loss increase $\Delta_j(X, Y)$ when removing feature $j$ from the dataset, cluster these local Importance vectors in $C$ groups, and then provide local feature importance to new points based on which cluster it is closest to.

The method is assessed on toy examples and 2 tabular datasets.

**Questions:**

## Questions
- In section 5.2, it is not clear how "product of feature values and their corresponding coefficients"
applies to the non-piecewise linear models (i.e. 2nd order models with and without interactions).
What measure of importance is used as ground truth for these two functions? This clarification is required for me to increase my score because the paper has a large emphasis on ground-truth evaluations.

- The model $f_{-j}$ used in R-LOCO is fitted on the whole dataset minus feature $j$. However, the proof of **Theorem 4.1** requires that the model $f_{-j}$ is **refitted** only using data that lands in a specific cluster $C_k$. This assumption is not part of the premise of the theorem nor is it discussed in the main text. It is only discussed on the caption of Figure 6 from the appendix. I think the premise of the theorem should be rewritten to include this assumption.

## Counter-example
I do not think that the R-LOCO approach is well motivated by focusing on piece-wise linear models. To prove my point, I propose a toy example where R-LOCO seemingly fails. I would like the authors to address or repudiate it.

Let $X_1,X_2\sim U(-1, 1)$ and $f(X_1, X_2) = X_2 \mathbb{1}(X_1\geq 0) - X_2\mathbb{1}(X_1< 0)$. From what I understand of the paper, an ideal regional explanation method should identify the two regions $\mathbb{1}(X_1<0)$ and $\mathbb{1}(X_1\geq 0)$ and report the linear model coefficient in each region (or at least feature importance consistent with these coefficients).
However, applying clustering in the feature importance space (instead of the input space) is not able to recover the two regions. Indeed, since the conditional expectation is the optimal predictor under the squared loss, we have
- $f_{-2}(x) = \mathbb{E}[f(X)|X_1=x_1] = 0$
- $f_{-1}(x)=\mathbb{E}[f(X)|X_2=x_2]=0$.

The corresponding LOCO effects using $Y=f(X)$ and the squared loss as the conformity score are

- $\Delta_1(X, f(X)) = (f(X)-f_{-1}(X))^2 = f(X)^2$
- $\Delta_2(X, f(X)) = (f(X)-f_{-2}(X))^2 = f(X)^2$

Now consider the map $\gamma: [-1, 1]^2 \rightarrow \mathbb{R}^2$ defined as $X\mapsto (\Delta_1(X, f(X)), \Delta_2(X, f(X)))=(f(X)^2, f(X)^2)$. Since $f(X_1, X_2) = -f(-X_1, X_2)$, we have that $\gamma(X_1, X_2) = \gamma(-X_1, X_2)$. That is, the two regions $\mathbb{1}(X_1<0)$ and $\mathbb{1}(X_1\geq 0)$ in input space completely overlap when mapped through $\gamma$. Consequently, no clustering algorithm applied in the LOCO space could recover the ground-truth regions.

Note that this counter-example is also applicable CSHAP (i.e. clustering on the L-SV). Therefore, this critique is not specific to R-LOCO but more to any clustering method that is applied on the feature importance space instead of the input space.

**Ethical Concerns:**

["NO or VERY MINOR ethics concerns only"]

**Final Justification:**

I have increased my score to "borderland accept" in light of the rebuttal.

**Limitations:**

The authors clearly highlight that the mathematical results all assume that the model is piece-wise linear. This assumption is not too restrictive seeing as it covers Tree-based methods (piece-wise constant) and ReLU networks.

**Paper Formatting Concerns:**

Figure 6 should be in the paper but I understand that it was moved to the appendix due to space constraints. Upon acceptance however, it should be put in section 4.

**Quality:**

3

**Strengths And Weaknesses:**

## Strengths

- The paper is well written and a breeze to read from start to finish.
- The technical sections 2, 3, and 4 are very clear.
- The limitation of SHAP highlighted by **Theorem 2.2** is an important contribution to the community.
- The experiments on toy data and models highlight the benefits of reporting regional explanations instead of fully local or global.

## Weaknesses
- Regional explainability is an emerging subject and there are existing methods that the current paper is not citing. For instance, the Effector package [1] (and the methods within) are not cited. Since [1] is still a preprint, I cannot be too severe on its exclusion in the experiments. Adding a citation + discussion would
be sufficient.

- The notion of "local feature importance" is not consistent within the paper. For instance, at line 123, it is stated that the feature local importance of a piece-wise linear model is the coefficient used in the region. At line 277, it is instead stated that the "product of feature values and their corresponding coefficients" is the local importance. These are two different importance measures and the correct one should be clarified directly after definition 2.1.

- Although I think the weakness of SHAP highlighted by **Theorem 2.2** motivates well the need for a regional explanation method, I do not think it motivates R-LOCO especially well. To explain why, I refer to the **Question** section below where I present a toy example where R-LOCO fails to recover ground-truth regions on a piece-wise linear model, and ask the authors to respond to this counter-example. This counter-example puts into question the emphasis on piece-wise linear models as a motivation for the approach.


[1] Effector: A Python package for regional explanations. https://arxiv.org/pdf/2404.02629v1

---

> ### Author Rebuttal · Authors · 2025-07-28
>
> We sincerely thank the reviewer for their thorough evaluation and insightful comments. This feedback is invaluable for improving the quality and clarity of our work. Below, we address each point in detail.
>
> ---
>
> > Q: On the Definition of Local Importance: how our definition of local importance: “the product of feature values and their corresponding coefficients”, applies to non-piecewise linear models (e.g., second-order models with or without interactions), and what measure of importance is used as ground truth for these models.
>
> **A**: The correct definition for our experiments is indeed the “product of feature values and their corresponding coefficients.” We will revise the text following Definition 2.1 to ensure this definition is applied consistently throughout the paper.
>
> For the non-piecewise linear models in Section 5.2, we define the ground-truth feature contribution using a standard functional decomposition, similar to a functional ANOVA.
>
> For a model $f(X)$ that can be decomposed into main effects and interactions:
>
> $f(X) = \sum_{i} f_i(X_i) + \sum_{i < j} f_{i,j}(X_i, X_j) + \dots,$
>
> the total contribution of a feature $X_k $ at a point $X$ is the sum of all components that include $X_k$. We will clarify this in the final version.
>
> For example, in our interaction model, the contributions are:
>
> $
> \text{Contribution of } X_1 = \text{Contribution of } X_2 = 3 \sqrt{3} \, X_1 X_2.
> $
>
> However, as we are not interested in directionality, we take the absolute value of the contributions for computing TP and FP.
>
> ---
>
> > Q: On the LOCO Model Fit within a Specific Cluster
>
> **A**: As noted in Figure 6 (where we provide details), once the regions are identified, a practitioner can apply any global importance method of choice to characterise them. In our experiments, the different approaches perform similarly. We plan to move Figure 6 into the main text and highlight this part more clearly.
>
> Regarding the proof of the theorem, while we presented it using regional estimators, we also noted that the version using global minimisers can be easily derived. In fact, using the global minimiser $\( f_{-j}(X) = \mathbb{E}[f(X) \mid X_{-j}] \)$ still satisfies the proof’s conditions, which requires that the estimator $f_{-j}(X)$ does not depend on variables outside the corresponding linear model used for the observation $X$. This condition is met whenever $X_j$ is locally important: removing this variable in the condition does not alter the switching variable, so both $f(x)$ and $f_{-j}(X)$ evaluate to the same region-specific affine function. We'll revise the explanation for clarity.
>
> ---
>
> > Q: On the Counter-Example
>
> **A**: We thank the reviewer for this excellent and insightful counter-example. The original counter-example uses the model:
>
> $
> f(X_1, X_2) = X_2 \cdot \text{sgn}(X_1)
> $
>
> with a squared-loss conformity score. This leads to an importance vector of: $(X_2^2, X_2^2)$ for all points, making the two regions $\( X_1 < 0 \)$ and $\( X_1 \ge 0 \)$ impossible to separate. This example underscores a key point we discuss in the paper (L389-391): the feature importance representation used for clustering is crucial, and motivates a promising direction for future work on enriching these representations to capture different facets of model behaviour.
>
> Before addressing this issue, we would like to emphasise that even in this example, if R-LOCO were to find an incorrect cluster or return just one cluster, the model would still avoid attributing importance to unused and independent variables, in contrast to SHAP, which is the main point of our paper.
>
> Now, regarding the importance of $X_2$. The clustering algorithm does not “see” \( X_1 \) at all; it only sees a single value $\( X_2^2 \)$ that determines the point’s position in the importance space. Consequently, the algorithm partitions data based on $\( X_2^2 \)$, and may forms clusters like:
>
> - **Cluster 1 (Low Importance):** Points where $\( X_2^2 \)$ is small (i.e., $\( X_2 \)$ is close to 0).
> - **Cluster 2 (High Importance):** Points where $\( X_2^2 \)$ is large (i.e., $\( X_2 \)$ is close to -1 or 1).
>
> This is still a valid description of the model’s behaviour with respect to $\( X_2 \)$, as it reveals regions where the magnitude of importance is homogeneous.
>
> ---
>
> #### **Adressing the Core Issue:** Enhanced Importance Vectors for Clustering
>
> The issue arises because squared loss is sign-agnostic: squaring discards the direction of the effect, which is crucial in distinguishing the two regions in the counter-example.
>
> As said in (L389-391), we could, for example, enrich the feature importance vectors to improve the clustering. In those lines, we suggested incorporating higher-order LOCO (e.g., removing subsets of features to compute LOCO). However, in this specific case, a more informative dimension would be the **signed effect**:
>
> $
> \delta_j(X) = f(X) - f_{-j}(X)
> $
>
> In the counter-example, this yields:
>
> $
> \delta_1^b(X) = \delta_2^b(X) = f(X)
> $
>
> So the full feature importance vector becomes $\gamma$:
>
> $$
> \gamma(X) = (f(X)^2, f(X)^2, f(X), f(X))
> $$
>
> - For $\( X_1 < 0 \): f(X) = -X_2  → (X_2^2, X_2^2, -X_2, -X_2) $
> - For $\( X_1 \ge 0 \): f(X) = X_2  → (X_2^2, X_2^2, X_2, X_2) $
>
>
> In this signed importance space, the two regions are now distinct and separable. For $\( X_2 \neq 0 \)$, points in $\( X_1 < 0 \)$ and $\( X_1 \ge 0 \)$ are mapped to opposite sides of the origin, allowing a standard clustering algorithm (e.g., K-Means) to easily distinguish the two groups.
>
> ---
>
> > Q: On Citation of [1]
>
> **A**: We thank the reviewer for bringing the Effector package [1] to our attention. This is indeed relevant prior work, and we will add an appropriate citation and discussion in the revised manuscript.

---

> ### Comment · Reviewer_v4Qd · 2025-08-04
> **Response to Rebuttal**
>
> I wish to thank the autors for responding to my concerns, especially to my counter example. I will increase my score to "borderline accept".
>
> I am still wondering how practical it is to augment the feature importance space with $\delta_j$ (and potentially feature interactions).  Like, what empirical criteria should be used to decide that the feature importance space should be augmented. For a toy example that is piecewise linear over regions, we can design the space so that the regions are separable in feature importance space. But what about real world data?

---

> ### Author Response · Authors · 2025-08-06
>
> Thank you for increasing your score.
>
> As outlined in the paper, the main goal of clustering in this context is to reduce the variance of feature attributions within each cluster. To achieve this, practitioners may choose to augment the feature importance representation, analogous to how feature engineering is used to improve predictive performance, guided by validation criteria. The key difference here is that the objective is not accuracy, but improved clustering quality in the importance space.
>
> Figure 3 highlights this effect: when comparing clustering in input space versus feature importance space, the **per-feature variance** is significantly lower in the latter. This supports our argument that clustering in the feature importance space leads to more coherent regions.
>
> We also plan to include your counterexample in the final version of the paper as a case study, to demonstrate when and how augmenting the space can be beneficial. Thank you again for this insightful contribution. We will include a note of acknowledgement to the anonymous reviewer in the final version.

---

> > ### Comment · Reviewer_v4Qd · 2025-08-06
> > **Reviewer Response**
> >
> > Thank you again for your clarification. Per-feature variance could be used as a criterion for deciding whether the feature importance space is expressive enough to recover the correct regions. But I think the impacts on the complexity of the regions $\hat{A}_k\subseteq \mathcal{X}$ in input space should also be considered.
> >
> > Indeed, as pointed out by **Reviewer nEVM**, the regions $\hat{A}_k$ can potentially be very complex and hard to interpret. This would result in feature importance measures that indeed have low variance, but which are only valid in regions that we cannot interpret. FDTrees [1] adress this concern by explicitelly defining the regions $\hat{A}_k$ via simple rules. One could do something similar with RLOCO and fit a classification tree to predict the cluster index of each data point. Nevertheless, I would expect the accuracy of said classification tree to degrade as we augment the feature importance space since the regions become more and more complex. Could the authors provide a general advice to practitionners in that regard?
> >
> > [1] Laberge, G., Pequignot, Y. B., Marchand, M., & Khomh, F. (2024, April). Tackling the xai disagreement problem with regional explanations. In International Conference on Artificial Intelligence and Statistics (pp. 2017-2025). PMLR.

---

> > > ### Author Response · Authors · 2025-08-06
> > >
> > > Thank you for this follow-up and for continuing the discussion.
> > >
> > > Regarding **Reviewer nEVM**’s comment, you’re absolutely right, and in fact, we implemented exactly the strategy you described: we fit a decision tree to predict the cluster labels in order to obtain an interpretable proxy for the regions. On our simulated example, this tree successfully recovers key information, such as the sign of the splitting variable. We refer you to our rebuttal for more details.
> > >
> > > In addition, our implementation allows users to supply their own interpretable clustering method, such as a decision tree directly, which enables reading region definitions in rule-based form. However, in our experiments, this approach typically performs slightly worse in terms of clustering quality for feature attributions compared to methods that focus purely on clustering performance, without prioritising interpretability.
> > >
> > > We also experimented with FDTree, which performed worse in our setup than purely clustering-driven approaches (see Figures 1 and 2).
> > >
> > > That said, we completely agree with your broader point: there is an inherent trade-off between interpretability and clustering quality for feature attributions, and this balance may depend on the complexity of the model being explained.
> > >
> > > At the same time, we see this concern as slightly beyond the primary scope of the paper. Our focus is on generating faithful feature attributions, which hinges on identifying coherent clusters. The interpretability of the *regions themselves*, while important, is a secondary layer of explanation. It’s a valuable direction for future work, and we appreciate you highlighting it.

---

### Official Review · Reviewer_GsiJ · 2025-06-19

**Clarity:** 3
**Significance:** 2
**Originality:** 2
**Rating:** 4
**Confidence:** 4

**Summary:**

This paper introduces a local importance method R-LOCO (Regional Leave Out Covariates), which tries to address limitations of SHAP and LIME. To improve the reliability of these methods, the authors first cluster the feature importance space, and then apply global importance methods to each cluster. The simulations and real data analysis shown the methods improved the previous local feature importance methods.

**Questions:**

1. In most of the literature, local importance typically refers to the importance assigned to a single prediction. In that sense, can R-LOCO be understood as calculating global importance for clusters?
2. The choice of the 10th and 95th percentiles as bound for the non-important features attributions in Section 5.1 is unconventional. Are there any considerations for not using 5th and 95th percentiles?
3. For the German credit datasets with 20 features, is there any reason only 19 features are shown in Figure 5 in Supplementary Material E?
4. Could you provide some intuition for why clustering in the feature importance space works better than clustering in input space as demonstrated in Section 5. Specifically, what causes the reduced variance of R-LOCO in Section 5.4?

**Ethical Concerns:**

["NO or VERY MINOR ethics concerns only"]

**Final Justification:**

All the issues are solved.

**Limitations:**

Yes.

**Quality:**

2

**Strengths And Weaknesses:**

Strength:

1. The paper identified some limitations in local importance methods in SHAP and LIME.
2. The paper proposes a method that combines clustering and LOCO to bridge local and global importance methods.
3. In simulation and real-world datasets, R-LOCO outperformed SHAP and LIME in terms of both TP and FP.

Weakness:

1. The authors only considered the case that the population can be clustered. However, it’s unclear how R-LOCO performs when the population is homogenous. In such cases, I would expect clustering to be less effective. I think it could be a good sensitive analysis, particularly if its performance remains comparable to Global LOCO. This could help demonstrate R-LOCO’s robustness.
2. The paper didn’t clearly state how the model determines which features are non-important. If the means are all non-zero as shown in Section 5.1, it’s confusing how are negatives identified. A clearer description of the decision rule would improve readability.
3. The “Mean Performance Drop” in Figure 1 and Figure 2 is negative, I believe it should be positive instead when important features are perturbed.
4. For better clarity, it’s better to at least mention the dimension of 2nd-order interaction model in Section 5.1.

Typos and formatting:

1. Missing period in line 377.
2. Figure 1 and Figure 2 which are associated with Section 6 are placed on page 7, but Figure 3 associated with Section 5.4 is placed on page 8. I think it could be rearranged for better readability.

---

> ### Author Rebuttal · Authors · 2025-07-28
>
> We thank the reviewer for their comments and the opportunity to clarify our contributions. Below, we address the specific points raised.
>
> ---
>
> > Q: Could you provide some intuition for why clustering in the feature importance space works better than clustering in input space as demonstrated in Section 5? Specifically, what causes the reduced variance of R-LOCO in Section 5.4?
>
> **A**: Consider the model
>
> $
> f(X) = (X_1 + X_2)\cdot 1_{X_6 \le 0} + (X_3 + X_4)\cdot 1_{X_6 > 0},
> $
>
> Clustering in the feature importance space (generated by R-LOCO with squared loss conformity score) yields a more robust and functionally meaningful separation of the model's two operational regions than clustering in the original input space. Below, we demonstrate this by analysing the decision rules associated with clustering in each space.
>
> #### Part 1: Clustering in Input Space
>
> The model defines two regions based on the value of $X_6$:
>
> - **Region A ($R_A$):** $\( X_6 \le 0 \)$
> - **Region B ($R_B$):** $\( X_6 > 0 \)$
>
> The effectiveness of clustering depends on the geometric separation between the point clouds of $R_A$ and $R_B$. We analyse this by computing their theoretical centroids:
>
> The centroid of Region A is
>
> $
> C_A = \mathbb{E}[X \mid X \in R_A] = (0, 0, 0, 0, 0, -0.5),
> $
>
> and the centroid of Region B is
> $C_B = \mathbb{E}[X \mid X \in R_B] = (0, 0, 0, 0, 0, 0.5)$.
>
> **Proof:**
> For $i \in \{1, ..., 5\}$, $X_i$ is independent of $X_6$, so
> $\( \mathbb{E}[X_i \mid X_6 \le 0] = \mathbb{E}[X_i] = 0 \)$.
> $\( X_6 \sim \text{Uniform}[-1, 0] \Rightarrow \mathbb{E}[X_6 \mid X_6 \le 0] = -0.5 \)$.
>
>
> **Conclusion for Part 1:**
>
> Although the centroids are distinct, the difference lies only in $X_6$, such that any variability in the other five dimensions $X_1,\dots, X_5$ introduces noise, making geometric separation in input space unreliable for clustering, as we can show by considering the decision rule:
>
> $x \in R_A$ if $d(x,C_A)^2-d(x,C_B)^2<0$.
>
> This rule is the one that underlies K-means clustering, where each observation is attributed to the region with the closest centroid. Expanding this gives:
>
> $d(x,C_A)^2-d(x,C_B)^2 = -2 \langle x, (C_A-C_B) \rangle + \Vert C_A \Vert^2 - \Vert C_B \Vert^2$,
>
> So the clustering decision rule reduces to
>
>  $x \in R_A \Leftrightarrow -2 <x,(C_A-C_B)> + \Vert C_A \Vert^2 - \Vert C_B \Vert^2 < 0 $ or $ \langle x, (C_A-C_B) \rangle > 0$
>
> This rule is similar to the classical linear SVM $sign(\langle x,(C_A-C_B) \rangle +b)$ (where +1 means $R_A$, and -1 means $R_B$) with normal vector $(C_A-C_B)=(0,0,0,0,0,-1)$. Thus, only $X_6$ is used for discriminating between the two groups (and roughly speaking $sign(X_6)$). Although the centroids are distinct in the $X_6$ dimension,  noise from the other five features (which are irrelevant to the decision boundary) hinders reliable clustering in input space. We show in part 2 that the clustering in R-LOCO Importance Space gives much better separation.
>
> ---
>
> #### Part 2: Clustering in R-LOCO Importance Space
> Each point $X$ is mapped to an importance vector
>
> $
> \gamma(X) = (\Delta_1, \dots, \Delta_6),
> $
>
> where   $\Delta_j(X) = (f(X) - f_{-j}(X))^2,$  and the LOCO predictor is $f_{-j}(X) = \mathbb{E}[f(X) \mid X_{-j}]$.
>
> For $X \in R_A$, the importance vector is
>
> $
> \gamma_A(X) = \left(X_1^2, X_2^2, 0, 0, 0, 0.25(X_1 + X_2 - X_3 - X_4)^2 \right).
> $
>
> ---
>
> **Proof:**
> In Region A, $f(X) = X_1 + X_2$.
>
> -  $f_{-1}(X) = \mathbb{E}[X_1 + X_2 \mid X_{2..6}] = \mathbb{E}[X_1] + X_2 = X_2 \Rightarrow \Delta_1 = X_1^2$
> - $f_{-2}(X) = \mathbb{E}[X_1 + X_2 \mid X_{1,3..6}] = X_1 + \mathbb{E}[X_2] = X_1 \Rightarrow \Delta_2 = X_2^2$
> - For $j \in \{3,4,5\}$, $\Delta_j = 0$
> - $f_{-6}(X) = 0.5(X_1 + X_2) + 0.5(X_3 + X_4) \Rightarrow$
>
>   $
>   \Delta_6 = \left(X_1 + X_2 - 0.5(X_1 + X_2 + X_3 + X_4) \right)^2 = 0.25(X_1 + X_2 - X_3 - X_4)^2
>   $
> ---
>
> As previously, we compute in the feature space the decision rule for clustering the feature vectors by computing the centroids.
>
> The centroid of Region A’s importance vectors is
>
> $
> C_{\gamma,A} = \left(\tfrac{1}{3}, \tfrac{1}{3}, 0, 0, 0, \tfrac{1}{3} \right).
> $
>
> ---
>
> **Proof:**
> Since all  $X_i \sim \text{Uniform}[-1,1]$:
> $\( \mathbb{E}[X_i^2] = 1/3 \)$, and all cross terms vanish due to independence and zero mean.
>
> ---
>
> By symmetry, for $\( X \in R_B \)$:
>
> $
> \gamma_B(X) = \left(0, 0, X_3^2, X_4^2, 0, 0.25(X_1 + X_2 - X_3 - X_4)^2 \right)
> $
>
> and the centroid is
>
> $
> C_{\gamma,B} = \left( 0, 0, \tfrac{1}{3}, \tfrac{1}{3}, 0, \tfrac{1}{3} \right).
> $
>
> ---
>
> **Conclusion for Part 2:**
>
> As before, the decision rule is
>
> $sign(<\gamma(x), (C_{\gamma,A}-C_{\gamma,B})> +b)$,
>
>  where +1 means $R_A$, and -1 means $R_B$ and
> $C_{\gamma,A}-C_{\gamma,B}= \left(\tfrac{1}{3}, \tfrac{1}{3}, -\tfrac{1}{3}, -\tfrac{1}{3}, 0 \right)$. Thus,
>
>  $x \in R_A$ if $sign(<\gamma(x), C_{\gamma,A}-C_{\gamma,B})>0$.
>
> The features used for each cluster $R_A, R_B$ are only based on the local important features, unlike input space clustering, the importance space reveals separation across multiple dimensions $\( \Delta_1, \Delta_2, \Delta_3, \Delta_4 \)$. More precisely,
>
> - **Region A**
>   If a point is in Region A, $f(\mathbf{X})$ depends on $\(X_{1}\)$ and $\(X_{2}\)$.
>   Thus, $\Delta_{1}$ and $\Delta_{2}$ will be large, while $\Delta_{3}$ and $\Delta_{4}$ will be zero.
>   The expression will be positive.
>
> - **Region B**
>   If a point is in Region B, $f(\mathbf{X})$ depends on $\(X_{3}\)$ and $\(X_{4}\)$.
>   Thus, $\Delta_{3}$ and $\Delta_{4}$ will be large, while $\Delta_{1}$ and $\Delta_{2}$ will be zero.
>   The expression will be negative.
>
> This captures fundamentally different patterns of feature usage, exactly what R-LOCO is designed to uncover.
>
> ---
>
> **Final Conclusion:**
> While clustering in input space gives only information on the globally important variable, R-LOCO reveals a multi-dimensional separation aligned with the model's logic. It transforms the problem from finding a geometric partition to identifying consistent feature relevance patterns.
>
> ---
>
> > Q: The paper didn’t clearly state how the model determines which features are non-important. If the means are all non-zero as shown in Section 5.1, it’s confusing how negatives are identified. A clearer description of the decision rule would improve readability.
>
> **A**: Lines 276–278 explain that we focus solely on variables directly involved in the computation of the final prediction after the condition. We use the absolute product of feature values and their corresponding coefficients as a measure of importance.
>
> We are not concerned with directionality here. Our primary test is whether a method assigns importance to independent and unused variables, not whether it captures sign.
>
> For example, for a model
>
> $
> f(X) = \sum_i f_i(X_i) + \sum_{i < j} f_{i,j}(X_i, X_j) + \dots,
> $
>
> the total contribution of a feature $\( X_k \)$ at point $\( X \)$ is the sum of all components involving $\( X_k \)$.  In our interaction model:
>
> $
> \text{Contribution of } X_1 = \text{Contribution of } X_2 = 3 \sqrt{3} X_1 X_2.
> $
>
> As we are not interested in directionality, we take the absolute value of the contributions for computing TP and FP.
>
> We will clarify this in the revision.
>
> ---
>
> > Q: The authors only considered cases where the population can be clustered. What happens when the population is homogeneous? Could a sensitivity analysis help validate R-LOCO’s robustness?
>
> **A**: Our main goal is to demonstrate a clear failure case for traditional baselines. Piecewise models are the most natural example where locality breaks down, hence our focus on them.
>
> However, our real-world example, where the population is likely not clusterable, shows that R-LOCO remains robust and effective compared to existing methods.
>
> ---
>
> > Q: Why use the 10th and 95th percentiles for non-important feature attributions in Section 5.1? Why not 5th and 95th?
>
> **A**: No particular reason, we verified that the conclusions hold using the 5th and 95th percentiles. We will update the manuscript accordingly.
>
> ---
>
> > Q: In the German Credit dataset (20 features), why are only 19 shown in Figure 5 (Supplementary Material E)?
>
> **A**: This was a mistake. The last feature was accidentally omitted while improving figure visibility in Inkscape.
>
> ---
>
> > Q: The “Mean Performance Drop” in Figures 1 and 2 is negative. Shouldn’t it be positive when important features are perturbed?
>
> **A**: Apologies for the confusion. We used the error, not accuracy. The base error is used as the baseline.
>
> ---
>
> > Q: For clarity, the dimension of the 2nd-order interaction model in Section 5.1 should be mentioned.
>
> **A**: We will include this detail. However, note that only the variables actually used in the model's definition are considered.
>
> ---
>
> > Q: In most of the literature, local importance typically refers to the importance assigned to a single prediction. In that sense, can R-LOCO be understood as calculating global importance for clusters?
>
> Yes.

---

> > ### Comment · Reviewer_GsiJ · 2025-08-08
> >
> > Thank you for your comments. I have raised my score to 4.

---

### Official Review · Reviewer_nEVM · 2025-07-01

**Clarity:** 2
**Significance:** 3
**Originality:** 3
**Rating:** 4
**Confidence:** 4

**Summary:**

The paper under review tackles the timely and important problem of providing principled, interpretable and trustworthy feature attributions for predictions of black-box models. The authors argue that the widely used methods LIME and Local Shapley Values (LSV) can assign importance to features that do not locally influence the model's output, even when the features are independent. Inspired by the fact that global importance methods like Leave Out COvariates (LOCO) do not suffer from this problem, they propose to provide local feature importance in the form of global feature importance scores for a restriction of the predictor to a specific region of the input space. They exploit the fact that the LOCO scores for a feature is formulated as  an expectation over inputs of the gap in performance with respect to a best predictor that does not have a access to that feature. They explore different clustering methods to identify regions that enjoy low variance in performance gaps, effectively leading to sets of inputs for which the expectation is a good approximation. They first evaluate their approach in a controlled setting where the data generating processes are explicitly defined by pieces, providing evidence that their approach succeeds at retrieving more accurately the variables that effectively appear on each piece while less often identifying variables that do not, compared to LIME and LSV. They further use their approach on two real world datasets to compare its fidelity towards observed performance drops with two other regional approaches.

**Questions:**

Could you provide examples of the regions identified by R-LOCO using AffinityPropagation and include a discussion on the interpretability vs "clustering quality" trade-off? Such a discussion should be included in subsection 5.4 in my opinion.

In section 6, it seems to me that R-LOCO and the two other baselines provide feature importance on each identified region. How were these multiple feature importance views used to obtain a single variable ranking used in Figure 1 and 2? If a single ranking is indeed used, why not compare to LOCO in this evaluation?

As a follow up question, what would be the importance ranking provided by R-LOCO with true clusters for the first order model used in the paper? Since $X_6$ has no importance on both regions would it be ranked after $X_1,\ldots, X_4$? If so, it seems that by chosing carefully the paramaters, the performance of $f_{-6}$ can be made arbitrarily low, which seems to me like an undesirable property.

In definition 2.1, is the whole function required to be continuous? While the name refer to $m$-continuity, no requirement is stated about this. Also this terminology ($m$-CPWL) is introduced, but no reference is made elsewhere in the paper if I'm not mistaken. What is the point of introducing this definition?

In the proof of Theorem 4.1, it is assumed that $\hat{f}_{0},\hat{f}_{j}$ are estimated for each region, which seems to contradict lines 191-192 that present them as estimators of population minimizers. Could you clarify this point? In your implementation, is the clustering done using local or global minimizers?

Line 215, is the boundary set $\hat{A}_{K+1}$ used in practice? What local attribution is computed in this case?

Line 247, what criterion is used for clustering in input space?

Line 281, It seems to me that this claim does not necessarily hold. Could you please elaborate on this or provide a reference for this fact?

Lines 277-278, could you provide formulas for the ground truth attributions you use, in particular for the model with interactions?

**Ethical Concerns:**

["NO or VERY MINOR ethics concerns only"]

**Final Justification:**

Good paper addressing important questions about explainability of black box models. The main issue of the proposed approach resides in the lack of interpretability of the regions which potentially hinders the overall usefulness of the final explanations.

**Limitations:**

Yes.

**Paper Formatting Concerns:**

None.

**Quality:**

3

**Strengths And Weaknesses:**

The paper is well written and proposes an interesting perspective on local feature importance by drawing inspiration from global feature importance. Experiments are interesting and well chosen to provide the reader with a good perspective on the challenges of feature attributions and the contributions of the paper.
In a sense, the method can be described as identifying a classifier $\kappa:\mathcal{X}\to [1,\ldots,m]$ and estimate the global variable importance scores LOCO $\Psi^k$ for the restrictions on the regions $A_k=\{x\in\mathcal{X} : \kappa(x)=k\}$. The local variable importance for an input $x$ is then given by $\Psi^{\kappa(x)}$. While $\Psi^k$ is well justified (at least in the case of independent features and in absence of interactions) for the predictor restricted on $A_k$, the method does not account for variable importance of the classifier $\kappa$. In extreme cases (for a fine enough partition), this approach would result in a constant predictor on each region (and so $\Psi^k=\vec{0}$ for all $k$) while the classifier $\kappa$ could be arbitrary complex. In this extreme case, R-LOCO would present null variable importance for all inputs, even though the predictor may be arbitrarily complex and actually rely on several specific variables, which can arguably be viewed as influencing the global model's output. I believe that the interpretability of the classifier $\kappa$, and thereby the complexity of the partition, is a key component of such an approach, which is not discussed in the paper. It would appear to me that any approach which relies on partition of the input space needs to deal with a trade-off. While partitions with a large number of small regions will lead to smaller variance, they are also harder to interpret.

Moreover, if a feature may not be important locally (like $X_6$ in the first order model), I would argue that it is an important feature when it comes to R-LOCO feature importance, since R-LOCO feature importance (say with true clusters) at input $X$ depends on the sign of $X_6$. I believe that in many applications this importance of $X_6$ cannot be overlooked. If the importance of the variables on which depends the clustering is viewed as a separate problem by the authors, I find that the paper lacks a discussion on how to reconcile this phenomenon with the local feature importance towards a global understanding of the model.

Related to the method and the evaluation, I find that some important information is missing (see questions) which makes it difficult to understand the advantages and limitations of the proposed approach.
In particular, it seems to me that the evaluation on real world datasets is focusing on global feature importance where as the proposed approach is specifically designed for local feature importance.

---

> ### Author Rebuttal · Authors · 2025-07-30
>
> We thank the reviewer for their comments and the opportunity to clarify our contributions. Below, we address the specific points raised.
>
> > Q: Could you provide examples of the regions identified by R-LOCO using AffinityPropagation and include a discussion on the interpretability vs "clustering quality" trade-off? Such a discussion should be included in subsection 5.4.
>
> **A**: Thank you for the suggestion. We agree that illustrating the identified regions will improve our analysis. However, most clustering algorithms only assign group labels without offering interpretability. In response, we fit a decision tree to predict the cluster labels, providing an interpretable proxy for the regions identified.
>
> Interestingly, we observed that R-LOCO correctly identifies the important regions by first splitting on the sign of X6, in contrast to clustering in the input space. This insight will be incorporated into Subsection 5.4 to better illustrate the interpretability vs clustering quality trade-off as suggested. See the regions found below:
>
> The region for R-LOCO in feature importance space is:
> ```
> |--- X6 <= 0.00
> |   |--- X5 <= -0.86
> |   |   |--- class: 0
> |   |--- X5 >  -0.86
> |   |   |--- class: 0
> |--- X6 >  0.00
> |   |--- X4 <= -0.56
> |   |   |--- class: 1
> |   |--- X4 >  -0.56
> |   |   |--- class: 0
> ```
> and the region for R-LOCO in input space is:
> ```
> |--- X5 <= -0.01
> |   |--- X2 <= -0.00
> |   |   |--- class: 0
> |   |--- X2 >  -0.00
> |   |   |--- class: 1
> |--- X5 >  -0.01
> |   |--- X6 <= 0.05
> |   |   |--- class: 2
> |   |--- X6 >  0.05
> |   |   |--- class: 3
> ```
> ---
>
> > Q: In Section 6, how were the multiple feature importance views combined into a single ranking for Figure 1 and 2? Why not include LOCO in the comparison?
>
> **A**: In this experiment, we were interested in comparing the “local” methods by perturbing each instance based on its absolute instance-wise importance. Since real-world data lacks ground-truth explanations, we rely on perturbation-based proxies that operate at the instance level. Using LOCO would result in the same perturbation across all instances, which does not align with the local evaluation setup.
>
> However, we agree it is a valid point, and we will include LOCO as a baseline in the revised version. In our experiments, it performed less well than R-LOCO.
>
> ---
>
> > Q: As a follow up question, what would be the importance ranking provided by R-LOCO with true clusters for the first order model used in the paper? Since $X_6$ has no importance on both regions would it be ranked after
> $X_1, \cdots, X_4$? If so, it seems that by chosing carefully the paramaters, the performance of
>  can be made arbitrarily low, which seems to me like an undesirable property.
>
> **A**: Comparing the importance of the splitting variable to other features is not straightforward. We excluded it from our evaluation to avoid introducing bias as the splitting variable can be viewed as globally important. Instead, we focus on variables that are either locally important (as per our definition) or not important at all (i.e., functionally and statistically independent).
>
> To clarify further, we conducted a more fine-grained analysis (See proof in rebuttal of reviewer GsiJ). For the first-order model, assuming $R_A = {X_6 \leq 0}$ and $R_B = {X_6 > 0}$ we explicitly computed the feature importance variable
>
> - For $\( X \in R_A \)$:
>   $$
>   \gamma_A(X) = \left(X_1^2, X_2^2, 0, 0, 0, 0.25(X_1 + X_2 - X_3 - X_4)^2 \right)
>   $$
> - For $\( X \in R_B \)$:
>   $$
>   \gamma_B(X) = \left(0, 0, X_3^2, X_4^2, 0, 0.25(X_1 + X_2 - X_3 - X_4)^2 \right)
>   $$
>
> Averaging these expressions over their respective regions yields equal importance for the relevant features in each region:
> - For $\( X \in R_A \)$:
>   $$
>   E_{R_A}[\gamma_A(X)] = \left(1/12,1/12, 0, 0, 0, 1/12 \right)
>   $$
> - For $\( X \in R_B \)$:
>   $$
>   E_{R_B}[\gamma_B(X)] = \left(0, 0, 1/12, 1/12, 0, 1/12 \right)
>   $$
>
>  We will include this analysis in the appendix.
>
> ---
>
> > Q: In Definition 2.1, is the function required to be continuous? Also, the term "-CPWL" is introduced but not used again.
>
> **A**: The definition was adapted from the cited papers. However, in our specific setting, we focus on each m function being affine. The definition was meant to introduce the class of piecewise linear functions used in the following theorem. We will revise the text
> to make it clearer and use the affine function instead of continuity. Thank you for pointing this out.
>
> ---
>
> > Q: In the proof of Theorem 4.1, it is assumed that $\hat{f}{0},\hat{f}{j}$ are estimated for each region, which seems to contradict lines 191-192 that present them as estimators of population minimizers. Could you clarify this point? In your implementation, is the clustering done using local or global minimizers?
>
> **A**: Thank you for pointing this out. Clustering is performed using global minimisers.
>
> As we mentioned in Figure 6 (which we will move into the main paper), once regions are identified, in principle, any global method can be applied within each region.
>
> The proof uses regional estimators for $f_{-j}(X)$. However, as we said in L567, the other case can be easily deduced. In fact, using the global minimiser $\( f_{-j}(X) = \mathbb{E}[f(X) \mid X_{-j}] \)$ still satisfies the proof’s conditions, which requires that it does not depend on variables outside the corresponding model used for the observation $X$. This condition is met whenever $X_j$ is locally important: removing this variable in the condition does not alter the switching variable, so both $f(x)$ and $f_{-j}(X)$ evaluate to the same region-specific affine function. We'll revise the explanation for clarity.
>
> ---
>
> > Q: Line 215 – Is the boundary set used in practice? What local attribution is computed there?
>
> **A**: As mentioned, the boundary set can be treated as a separate group. However, such cases are very rare in practice. Typically, ties can be broken by randomly assigning the sample to one of the valid regions. We will clarify this in the final version.
>
> ---
>
> > Q: Line 247 – What criterion is used for clustering in input space?
>
> **A**: We use the same clustering algorithm and criterion as for the feature importance. We just swap the input matrix with the feature importance matrix.
>
> ---
>
> > Q: Line 281 – The claim seems questionable (LOCO values being between (0, 1). Could you elaborate or provide a reference?
>
> **A**: Thank you for pointing this out. LOCO is often linked to the variance explained (e.g., Sobol indices); its values are positive, and when properly normalised, typically fall within a defined range. We will clarify this and cite sources such as: "Feature Importance: A Closer Look at Shapley Values and LOCO."
>
> ---
>
> > Q: Lines 277–278 – Could you provide the formula for ground-truth attributions, especially for the interaction model?
>
> **A**: Yes, we use the absolute product of feature values and their corresponding coefficients as the ground-truth measure of importance. We do not account for directionality in this evaluation.
>
> For example, for a model
>
> $
> f(X) = \sum_i f_i(X_i) + \sum_{i < j} f_{i,j}(X_i, X_j) + \dots,
> $
>
> the total contribution of feature $\( X_k \)$ at point $\( X \)$ is the sum of all components involving \( X_k \). In our interaction model:
>
> $
> \text{Contribution of } X_1 = \text{Contribution of } X_2 = 3 \sqrt{3} X_1 X_2
>
> We compute true/false positives using the absolute value of the contributions.

---

> > ### Comment · Reviewer_nEVM · 2025-08-06
> > **figures 1and 2**
> >
> > Thank you for your reply. I'm still missing the precise steps leading to Figures 1 and 2. I apologize if this information is present in the paper. Given that these results are the main basis for the superiority of the proposed approach, I think this is important.

---

> > > ### Author Response · Authors · 2025-08-06
> > >
> > > Thank you for your comment. We will improve the explanation in Lines 366–369 by incorporating a formal definition of the metrics used to generate Figures 1 and 2, as detailed below:
> > >
> > > ---
> > >
> > > ### **Definitions**
> > >
> > > We begin by outlining the core components of our evaluation framework:
> > >
> > > * **Dataset `D`**: A collection of `n` samples. Each sample is a pair `(x, y)`, where `x` is the input and `y` is the ground-truth target.
> > > * **Input `x`**: A list of `d` numerical features.
> > > * **Model `f(x)`**: The predictive model that maps input `x` to a prediction.
> > > * **Attribution Method `A(x)`**: A function that assigns a positive importance score to each of the `d` input features.  These scores correspond to the normalised attributions from each method (e.g., R-LOCO, SHAP).
> > > * **Error Function `E(x, y)`**: A metric that quantifies the discrepancy between the model’s prediction for `x` and the true label `y`:
> > >
> > >   * **Classification**: `E(x, y) = 1` if the prediction is incorrect, `0` otherwise.
> > >   * **Regression**: `E(x, y)` may be the absolute difference between the predicted and true values.
> > >
> > > ---
> > >
> > > ### **Masking Procedure**
> > >
> > > To assess feature importance, we apply a **masking** (or **occlusion**) technique:
> > >
> > > 1. For each input `x`, compute the importance scores using `A(x)`.
> > > 2. Identify the indices of:
> > >
> > >    * the top-`k` most important features (**Top-k Indices**),
> > >    * and the bottom-`k` least important features (**Bottom-k Indices**).
> > > 3. Create two modified versions of the input:
> > >
> > >    * `x_masked_top_k`: Replace values at Top-k Indices with zero.
> > >    * `x_masked_bottom_k`: Replace values at Bottom-k Indices with zero.
> > >
> > > ---
> > >
> > > ### **Error Calculation**
> > >
> > > We then compute average model errors over the dataset under three conditions:
> > >
> > > * **Original Average Error (`AvgError_Original`)**:
> > >
> > >   $$
> > >   AvgError\\\_Original = \frac{1}{n} \sum_{(x, y) \in D} E(x, y)
> > >   $$
> > >
> > > * **Top-k Masked Average Error (`AvgError_Top_k`)**:
> > >
> > >   $$
> > >   AvgError\\\_Top\\\_k = \frac{1}{n} \sum_{(x, y) \in D} E(x_{masked\\\_top\\\_k}, y)
> > >   $$
> > >
> > > * **Bottom-k Masked Average Error (`AvgError_Bottom_k`)**:
> > >
> > >   $$
> > >   AvgError\\\_Bottom\\\_k = \frac{1}{n} \sum_{(x, y) \in D} E(x_{masked\\\_bottom\\\_k}, y)
> > >   $$
> > >
> > > ---
> > >
> > > ### **Final Metrics for Figures**
> > >
> > > The values shown in Figures 1 and 2 represent **the change in model error** due to masking, computed as follows:
> > >
> > > * **Top-k Error Change (Left figure)**:
> > >   Indicates the degradation in performance after removing the Top-*k* most important features:
> > >
> > >   $$
> > >   ChangeInError\\\_Top(k) = AvgError\\\_Original - AvgError\\\_Top\\\_k
> > >   $$
> > >
> > > * **Bottom-k Error Change (Right figure)**:
> > >   Reflects the performance change after removing the Top-*k* least important features:
> > >
> > >   $$
> > >   ChangeInError\\\_Bottom(k) = AvgError\\\_Original - AvgError\\\_Bottom\\\_k
> > >   $$
> > >
> > > ---
> > >
> > > We hope this clarifies the evaluation methodology. This detailed explanation will be incorporated into the revised version of the paper.
> > >
> > > ---

---

> > > > ### Comment · Reviewer_nEVM · 2025-08-07
> > > >
> > > > Thank you for this detailed explanation about the procedure for producing Figures 1 and 2. How do you justify the occlusion by replacing feature values by the specific value zero? Are the numerical features centered? How does this specific choice affect the result?

---

> > > > > ### Author Response · Authors · 2025-08-07
> > > > >
> > > > > Thank you for your question. Occluding or removing features by setting them to a baseline value is a common approach for evaluating feature attribution in explainable AI (e.g., Hooker et al., Evaluating Feature Importance Estimates, 2018). In our case, the features were normalised to have zero mean and unit variance. As a result, replacing a feature value with zero effectively resets it to the mean, a neutral baseline, thereby simulating the absence of information from that feature.
> > > > >
> > > > > We also experimented with occluding features using random noise and observed similar results, which we plan to report in the appendix.

---

> > > > > > ### Comment · Reviewer_nEVM · 2025-08-08
> > > > > >
> > > > > > Thank you for this answer which addresses several of my concerns. I have therefore raised my score to 4.

---

### Official Review · Reviewer_dyeU · 2025-07-02

**Clarity:** 3
**Significance:** 3
**Originality:** 3
**Rating:** 4
**Confidence:** 3

**Summary:**

The paper identifies flaws in commonly used local explanation methods: Local Shapley Values (L-SV) and LIME, even under idealized assumptions of independent features and exact computation. The authors prove via Theorem 2 that L-SV can assign nonzero importance to features outside the true active region of a piecewise-linear model, and similarly show LIME failures under the same setup. To remedy this, they propose R-LOCO (Regional Leave Out COVariates), which:

1. Computes global LOCO importance for each observation.
2. Embeds instances in the “feature-importance space” defined by their LOCO vectors.
3. Clusters these embeddings into K sets (variants in table are based on different clustering schemes).
4. Assigns each point the region’s averaged LOCO scores as its local explanation.

They prove in Theorem 4.1 that if a cluster lies entirely within a true region, R-LOCO recovers exact local attributions for piecewise-linear models. Through controlled simulations with ground-truth explanations (first and second-order additive and interaction models), R-LOCO achieves true-positive rates up to 97–98% with false-positive rates below 4%, outperforming L-SV and LIME (TP 30–66%, FP 34–70%). In real-world tests on the Diabetes and California datasets (using XGBoost), R-LOCO also yields higher fidelity: masking its top-k features drops model performance more and masking bottom-k features drops it less, compared to L-SV, LIME, CSHAP, and the FDT baseline.

**Questions:**

*Scalability & Runtime*

1. Issue: LOCO’s leave one covariate approach requires fitting p + 1 models per cluster (or approximations thereof).
2. Request: Please report wall clock training and inference time (and memory usage) for R-LOCO vs. L-SV/LIME on the Diabetes and California datasets and discuss scaling behavior as p grows.

*Automatic Region Validation*

1. Issue: Cluster purity is critical for correctness (Theorem 4.1) but cluster-straddling degrades performance (App C).
2. Request: Please integrate a purity metric (e.g., variance threshold on LOCO vectors) to automatically detect and refine impure clusters. The goal is to answer: under what quantitative purity threshold does performance sharply degrade?

*High-Dimensional / Correlated Features*

1. Issue: All controlled experiments assume independent features. As the authors acknowledge real datasets often exhibit strong correlations. I am unsure how much of this correlation is depicted in the 2 datasets considered.
2. Request: Have you tested R-LOCO on synthetic correlated feature settings (e.g., Gaussian multivariate with block covariance) to gauge robustness? How do correlations affect clustering fidelity and TP/FP rates?

*Choice of Distance Metric*

1. Issue: Euclidean distance in LOCO space may underperform when features have different variance scales.
2. Question: This maybe another issue for high dimensional data. Have you compared Mahalanobis or cosine distances for clustering? Could adaptive metric learning improve region coherence?

*Applicability*

1. Issue: It remains unclear how R-LOCO generalizes to unstructured data or non-additive model parts (e.g., CNN feature maps).
2. Question: Are there specific use cases/applications that benefit from the computational overhead of the proposed method?

**Ethical Concerns:**

["NO or VERY MINOR ethics concerns only"]

**Final Justification:**

I am increasing my score to 4 and my confidence to 3. The additional results are along expected lines with the proposed method suffering due to scaling the number of features. However, for some applications (as noted by the authors in the rebuttal) it might be justifiable. The lack of more complicated synthetic settings in Section 5 is still a concern.

**Limitations:**

The paper does not explicitly discuss limitations in the main paper. However, there is an answer regarding limitations in the checklist. I recommend adding one in the main paper that discusses:
    • Clustering assumptions: Requirement that clusters align with true functional regions; sensitivity to hyperparameters.
    • Computational overhead: Multiple model fits for LOCO and clustering steps.
    • Model/domain scope: Currently validated on tabular, piecewise-linear or tree-based models; adaptation to deep networks or non-tabular data is future work.

**Quality:**

2

**Strengths And Weaknesses:**

**Quality**

*Strengths:*

1. Theoretical rigor: Full proofs for Theorems 2.2 and 4.1 appear in Appendices A and B, with clear statement of assumptions (piecewise linearity, independence, clustering inclusion).
2. Comprehensive experiments: Controlled benchmarks include 50 runs per setting, error quantiles for non-important features, and ablations over four clustering algorithms and hyperparameters (K = 2,4,8,20; damping = 0.5,0.6,0.9) demonstrating stability.
3. Real-world validation: Utilizes public datasets (Diabetes, California), high-performing XGBoost models, and two separate fidelity metrics (top-k and bottom-k masking).

*Weaknesses:*

1. Clustering reliance: Performance depends heavily on cluster purity (Appendix C). It is unclear how the method performs in in more high-dimensional data where feature overlap is higher. I was unable to find if automated purity checks are utilized in the paper
2. Compute cost: LOCO requires re-fitting models per region; no detailed runtime or memory analysis is given, limiting assessment of scalability to high-dimensional or large-scale data.
3. Applicability: LIME and shap values are applicable across domains (extensions to large-scale vision datasets as well). Due to reliance on clustering, I am unsure of the proposed method's applicability.

**Clarity**

*Strengths:*

1. Logical flow: Paper moves cleanly from theoretical critique (Sec. 2–4) to controlled experiments (Sec. 5) and real data (Sec. 6)
2. Notation consistency: Feature sets, clusters, and Shapley/LOCO notation remain consistent throughout.

*Weaknesses:*

1. Dense appendices: Key implementation details (e.g., distance metrics, damping choices) are relegated to appendices, forcing readers to flip back and forth. Fig 6 (which is in appendix) should be in the main paper.
2. Acronym overload: Readers must track R-LOCO, R-LOCOTC, R-LOCOIC, CSHAP, FDT consider consolidating or emphasizing defaults.

**Significance**

*Strengths:*

1. Bridges local/global gap: R-LOCO offers a general recipe to turn any global explainer into a regional/local one, addressing a central challenge in XAI.
2. Actionable insights: Demonstrates that clustering in importance space (vs. input space) yields markedly better fidelity, guiding future explainer design.

*Weaknesses:*

1. Domain scope: Focuses exclusively on tabular, piecewise-linear, or tree-based models; applicability to vision, NLP, or deep nets remains unclear.

**Originality**

*Strengths:*

1. Novel theoretical critique: Per my awareness, the paper is the first to formally prove failures of L-SV under ideal conditions for piecewise models.

*Weaknesses:*

1. Building blocks: Clustering algorithms and LOCO are existing techniques; R-LOCO’s novelty lies in their combination rather than entirely new algorithms.

---

> ### Author Rebuttal · Authors · 2025-07-30
>
> We thank the reviewer for their comments and the opportunity to clarify our contributions. Below, we address the specific points raised.
>
> > Q: Request: Please report wall clock training and inference time (and memory usage) for R-LOCO vs. L-SV/LIME on the Diabetes and California datasets and discuss scaling behavior as *p* grows.
>
> **A**: Thank you for this helpful suggestion. We have now conducted a detailed runtime and memory usage analysis for R-LOCO, LIME, and SHAP on both the California and Diabetes datasets.
>
> We find that R-LOCO is significantly faster at inference time than the baselines. While our method incurs additional cost during training, due to clustering and model fitting, it performs inference with minimal overhead, as only the cluster assignment is required per sample. We observe a similar pattern for memory usage.
>
> In terms of scalability, R-LOCO’s  **training** computational cost grows approximately linearly with the number of features. This behavior is on par with widely-used approaches such as SHAPLoss and variants (Explaining by Removing: A Unified Framework for Model Explanation). Note that it needs to be done only once. Furthermore, R-LOCO can be optimised for tree-based models using approaches such as Projected Forests (SHAFF: Fast and consistent SHApley eFfect estimates via random Forests), where a single model can be reused to efficiently compute multiple conditional expectations.
>
> We believe these additions substantially strengthen the contribution of our paper.
>
> #### California Dataset
> | Method   | Train (s) | Infer (s) | Total (s) | Train (MB) | Infer (MB) | Peak (MB) |
> |----------|-----------|-----------|-----------|-------------|-------------|-----------|
> | R-LOCO   | 33.521    | 0.032     | 33.553    | 1687.9      | 8.6         | 2743.0    |
> | LIME     | 0.038     | 34.684    | 34.722    | 0.5         | 2.3         | 1.0       |
> | SHAP     | 0.002     | 20.698    | 20.700    | 0.0         | 0.8         | 6.8       |
>
> #### Diabetes Dataset
>
> | Method   | Train (s) | Infer (s) | Total (s) | Train (MB) | Infer (MB) | Peak (MB) |
> |----------|-----------|-----------|-----------|-------------|-------------|-----------|
> | R-LOCO   | 1.611     | 0.010     | 1.621     | 12.2        | 0.5         | 1.7       |
> | SHAP     | 0.002     | 1.433     | 1.436     | 0.0         | 145.0       | 16.7      |
> | LIME     | 0.008     | 0.838     | 0.846     | 0.2         | 1.3         | 0.4       |
>
> ---
>
> > Q: Please integrate a purity metric (e.g., variance threshold on LOCO vectors) to automatically detect and refine impure clusters. The goal is to answer: under what quantitative purity threshold does performance sharply degrade?
>
> **A**: Thank you for the suggestion. While we report results with several clustering parameters to give a detailed analysis, in our current implementation, we include a knee-point strategy to choose the number of clusters. We also support any clustering class with `fit` and `predict` methods, enabling integration with methods that include automatic cluster validation.
>
> ---
>
> > Q: Have you tested R-LOCO on synthetic correlated feature settings (e.g., Gaussian multivariate with block covariance) to gauge robustness? How do correlations affect clustering fidelity and TP/FP rates?
>
> **A**: We have deliberately avoided introducing correlation in the synthetic experiments because our goal is to evaluate feature importance under the assumption of independence, where ground truth importance can be unambiguously defined.
>
> In correlated settings, disentangling the effects of statistical dependence from true functional importance becomes challenging, making it difficult to compute TP/FP rates in a principled way.
>
> That said, our experiments on real-world datasets inherently include correlations, and we evaluate performance there using perturbation-based evaluation as there is no ground truth available in the real-world setting.
>
> ---
>
> > Q: Have you compared Mahalanobis or cosine distances for clustering? Could adaptive metric learning improve region coherence?
>
> **A**:  As noted in lines 210–212 of the main paper, R-LOCO supports any similarity function during inference, including those used by the clustering algorithm. We have experimented with cosine distances in place of the Mahalanobis, and found minimal impact on overall conclusions.
>
> Nonetheless, we agree that in some settings, particularly in domain-specific structures, careful metric selection could yield improvements. Our implementation supports passing a custom distance function and clustering algorithm, giving users full control over this aspect.
>
> ---
>
> > Q: Are there specific use cases/applications that benefit from the computational overhead of the proposed method?
>
> **A**:  A significant majority of real-world machine learning applications, particularly in critical sectors like finance, healthcare, retail, and insurance, are in majority built on structured, tabular data. Many applications on tabular data can afford moderate computational overhead in exchange for more faithful and locally accurate explanations.
>
> Moreover, methods with similar computational profiles, such as SHAPLoss and variants (see *Explaining by Removing* paper) are already used in practice. R-LOCO offers additional interpretability benefits by structuring explanations into coherent regions, which is often desirable in real-world workflows where more localised decisions are important.
>
> ---
>
> > Q: However, there is an answer regarding limitations in the checklist. I recommend adding one in the main paper that discusses: • Clustering assumptions: Requirement that clusters align with true functional regions; sensitivity to hyperparameters. • Computational overhead: Multiple model fits for LOCO and clustering steps. • Model/domain scope: Currently validated on tabular, piecewise-linear or tree-based models; adaptation to deep networks or non-tabular data is future work.
>
> **A**:  Thank you for this important suggestion. We agree that these limitations should be more explicitly stated in the main paper and will revise accordingly.
>
> As noted in lines 227–229, we provide a theoretical analysis of the impact of imperfect or contaminated clusters on the fidelity of R-LOCO explanations. Additionally, Section 5.4 and Table 2 present an empirical analysis of how clustering affects performance.
>
> While R-LOCO requires model refitting per feature, similar to methods like SHAPLoss, this overhead is incurred only during training. Inference is highly efficient. Moreover, for tree-based models (commonly used in tabular data), the training cost can be significantly reduced using techniques such as Projected Forests (SHAFF: Fast and consistent SHApley eFfect estimates via random Forests).
>
> Indeed, our method is currently validated on tabular data, as are most feature removal-based explanations. Extending R-LOCO to deep learning models or non-tabular modalities is a promising direction for future work.
>
> We will make these limitations more explicit in the final version of the paper to ensure clarity and transparency.

---

> > ### Comment · Reviewer_dyeU · 2025-08-06
> > **Thanks for the detailed reply**
> >
> > I thank the authors for their detailed reply. The insights in Section 2 are significant. As the authors note, the additional computations are justified. I am increasing my score to 4 (Borderline Accept).
> >
> > I think there is an additional missing piece of analysis between the simplistic assumptions in Section 5 and real-world datasets in Section 6. Some form of Gaussian correlation models/cluster analyses to bridge this gap can be very helpful in understanding the results.

---

> > > ### Author Response · Authors · 2025-08-09
> > >
> > > Thank you for increasing your score. We appreciate your valuable feedback and suggestions, which will help us improve the paper.

---

### Decision · Program_Chairs · 2025-09-17

**Decision:**

Accept (poster)

**Comment:**

This submission first identifies failures of two popular feature attribution methods, Local Shapley Values (L-SV) and LIME, in finding locally important features. Motivated by these failures, it then proposes R-LOCO, a hybrid local/global explanation method that partitions the feature space into regions with homogeneous feature importance characteristics and then applies a global attribution method to each region.

Strengths identified by the reviewers are as follows:
- Valuable identification of failures of L-SV and LIME, for even a piecewise-linear model.
- R-LOCO bridges local and global feature importance, providing an interesting perspective on the former by drawing on methods for the latter.
- R-LOCO demonstrates higher fidelity than L-SV, LIME, etc. on real data.
- The paper is well-written and easy to read.

Weaknesses:
- Reviewers were concerned that the regions produced by R-LOCO may be complex and uninterpretable, hindering overall interpretability. While reviewers mostly accepted the argument in the author rebuttal that the complexity of regions is a problem for future work, they feel (and I support them as area chair) that the issue should be discussed more in the current paper. Specifically:
    - The number of clusters appears to be the main way to control region complexity.
    - In their rebuttal to Reviewer nEVM, the authors showed a decision tree fit to approximate the clusters/regions. This tree as well as additional trees should be included in the paper along with the accuracies of approximation, perhaps in a trade-off with the number of clusters.
    - The role of the feature importance space (as shown in the rebuttal to Reviewer v4Qd's counterexample) should also be discussed.
- Computational cost: R-LOCO requires fitting $O(p)$ models where $p$ is the number of features. The author rebuttal shows that this training cost is mitigated because inference is fast, having only to assign points to clusters. The rebuttal (to a different point, see below) also suggests that one only needs to fit $p$ models using the whole dataset and not for each cluster, which would further mitigate the concern.
- There was confusion about important points such as the definition of "local feature importance" and whether feature importance is estimated using regional (per cluster) or global estimators (using the whole dataset), among other clarifications and corrections. The rebuttal seems to have addressed these and the clarifications should of course be added to the paper. The region vs. global estimator question is particularly important for the point about computational cost.

All reviewers became positive toward the submission during the discussion period and agreed that it should be accepted. I support this consensus. At the same time, I emphasize that this consensus is dependent on the resolution of some concerns by the rebuttal and the corresponding recommended changes, as summarized above.